Manuscript prepared for J. Name
with version 3.2 of the LATEX class copernicus.cls.
Date: 22 April 2019

# Ice Nucleating Particles in a Coastal Tropical Site

Luis A. Ladino[1,*], Graciela B. Raga[1], Harry Alvarez-Ospína[2], Manuel A. Andino-Enríquez[3], Irma Rosas[1], Leticia Martínez[1], Eva Salinas[1], Javier Miranda[4], Zyanya Ramírez-Díaz[1], Bernardo Figueroa[5], Cedric Chou[6], Allan K. Bertram[6], Erika T. Quintana[7], Luis A. Maldonado[8], Agustín García-Reynoso[1], Meng Si[6], and Victoria E. Irish[6]

[1]Centro de Ciencias de la Atmosfera, Universidad Nacional Autonoma de Mexico, Mexico City, Mexico
[2]Facultad de Ciencias, Universidad Nacional Autonoma de Mexico, Mexico City, Mexico
[3]School of Chemical Sciences and Engineering, Universidad Yachay Tech, Ecuador
[4]Instituto de Fisica, Universidad Nacional Autonoma de Mexico, Mexico City, Mexico
[5]Laboratorio de Ingenieria y Procesos Costeros, Instituto de Ingenieria, Universidad Nacional Autonoma de Mexico, Sisal, Yucatan, Mexico
[6]Chemistry Department, University of British Columbia, Vancouver, Canada.
[7]Escuela Nacional de Ciencias Biologicas, Instituto Politecnico Nacional, Mexico City, Mexico
[8]Facultad de Quimica, Universidad Nacional Autonoma de Mexico, Mexico City, Mexico

*Correspondence to:* Luis A. Ladino (luis.ladino@atmosfera.unam.mx)

**Abstract.** Atmospheric aerosol particles that can nucleate ice are referred to as ice nucleating particles (INPs). Recent studies have confirmed that aerosol particles emitted by oceans can act as INPs. This very relevant information can be included in climate and weather models to predict the forma-

tion of ice in clouds, given that most of them do not consider oceans as a source of INPs. Very few studies to sample INPs have been carried out in tropical latitudes, and there is a need to evaluate their availability to understand the potential role that marine aerosol may play in the hydrological cycle of tropical regions.

This study presents results from the first measurements obtained during a field campaign con-

ducted in the tropical village of Sisal, located on the coast of the Gulf of Mexico of the Yucatan peninsula in Mexico in January-February 2017, and one of the few datasets currently available at such latitudes (i.e., 21°N). Aerosol particles sampled in Sisal are shown to be very efficient INPs in the immersion freezing mode, with onset freezing temperatures in some cases as high as -3 °C, similar to the onset temperature from *Pseudomonas syringae*. The results show that the INP concen-

tration in Sisal was higher than at other locations sampled with the same type of INP counter. Air masses arriving in Sisal after the passage of cold fronts have, surprisingly, higher INP concentrations than the campaign-average, despite their lower total aerosol concentration.

The high concentrations of INPs at warmer ice nucleation temperatures (T > -15 °C) and the supermicron size of the INPs suggest that biological particles may have been a significant contributor

to the INP population in Sisal during this study. However, our observations also suggest that at temperatures ranging between -20 °C and -30 °C mineral dust particles are the likely source of the measured INPs.

## 1   Introduction

Clouds are essential to the hydrological cycle of the planet and also play a significant role in the
radiative balance of the climate system (Ramanathan et al., 1989; Lohmann and Feichter, 2005; Andreae and Rosenfeld, 2008; Stevens and Feingold, 2009). Cloud formation depends on the presence of cloud condensation nuclei (CCN) and most precipitation from mixed-phase clouds involves also the presence of ice nucleating particles (INPs). Aerosol-cloud interactions are one of the main sources of uncertainty in climate projections as assessed by the Intergovernmental Panel on Climate
Change (Stocker et al., 2013), prompting large research efforts from the scientific community in recent years. Nevertheless, the formation and evolution of ice crystals in mixed-phase and cirrus clouds still remain highly uncertain (Seinfeld et al., 2016; Kanji et al., 2017; Field et al., 2017). Several pathways have been proposed as potentially responsible for ice formation: condensation freezing, contact freezing, immersion freezing, and deposition nucleation (Vali et al., 2015). Murray
et al. (2012) and Ladino et al. (2013) have suggested that contact freezing and immersion freezing are the most efficient mechanisms leading to ice nucleation in clouds; however, the atmospheric relevance of contact freezing is still unclear given the contradictory results (Hobbs and Atkinson, 1976; Ansmann et al., 2005; Cui et al., 2006; Phillips et al., 2007; Seifert et al., 2011; Kanji et al., 2017).

Most of the precipitation from deep convection in the tropics, e.g. in the Inter-tropical Conver-
gence Zone, forms via the ice phase (Mülmenstädt et al., 2015). Given the ice-nucleating potential of a variety of aerosol particles such as mineral dust, biological particles, crystalline salts, carbonaceous particles, and secondary organic aerosol, the main source of INPs at tropical latitudes is highly uncertain (Kanji et al., 2017; Yakobi-Hancock et al., 2014; DeMott et al., 2010). Although it is yet not fully understood what exactly makes an aerosol particle an efficient INP (e.g. its composition,
active sites, crystal structure, size, or hygroscopicity), there is evidence that their composition is one of the key factors (Kanji et al., 2017). On a global scale, the large tropospheric concentrations and the good ice nucleating abilities of mineral dust have been widely reported as an important INP source (Hoose and Möhler, 2012; Nenes et al., 2014; Atkinson et al., 2013; Kanji et al., 2017). Bioaerosol has also been identified as very efficient INP (Kanji et al., 2017; Hoose and Möhler,
2012; Fröhlich-Nowoisky et al., 2016; Hill et al., 2017), with onset freezing temperatures reported as high as -2 °C (Yankofsky et al., 1981; Després et al., 2012; Fröhlich-Nowoisky et al., 2015; Wex et al., 2015; Stopelli et al., 2017). Global climate models parameterize cloud droplet and ice crystal formation from observational studies and results from such modelling suggest that on a global scale bioaerosol is not a major source of INPs and therefore, have a lower impact in ice cloud formation

in comparison to mineral dust particles (Hoose et al., 2010; Sesartic et al., 2012). However, this may not be the case on a regional scale (Burrows et al., 2013; Mason et al., 2015a). Marine organic matter, likely of biological origin, has been suggested to be an important oceanic source of INPs in the southern oceans, north Atlantic, and north Pacific (Burrows et al., 2013; Yun and Penner, 2013; Wilson et al., 2015; Vergara-Temprado et al., 2017). However, the maritime source suggestion was

made with little or no data from tropical latitudes.

Important efforts were made during the 1950-70s to understand the role of the oceans in ice cloud formation (Bigg, 1973; Schnell and Vali, 1975; Schnell, 1975, 1977, 1982; Rosinski et al., 1987, 1988). There is recent new and robust evidence that biological material from the marine environment could act as efficient INPs (Knopf et al., 2011; Wilson et al., 2015; Mason et al., 2015b; DeMott

et al., 2016; Ladino et al., 2016; McCluskey et al., 2017; Irish et al., 2017; Welti et al., 2018). Most of the past available INP data were obtained from mid- and high-latitudes studies, with tropical latitudes heavily under represented (Schnell, 1982; Rosinski et al., 1987, 1988; Boose et al., 2016; Welti et al., 2018; Price et al., 2018). Marine and coastal INP concentration ([INP]) typically ranges from $10^{-4}$ L$^{-1}$ to $10^{-1}$ L$^{-1}$ for temperatures between -10 °C and -25 °C (Kanji et al., 2017), but

have shown to be higher at tropical coastal sites (Rosinski et al., 1988; Boose et al., 2016; Welti et al., 2018; Price et al., 2018). This large [INP] range may strongly depend on the microbiota concentration, the marine biological activity, and the organic matter enrichment in the sea surface microlayer as shown in Wilson et al. (2015).

At marine and coastal sites, a large variety of bacteria have been identified with *Proteobacteria*,

*Firmicutes*, and *Bacteroidetes* as the main reported phyla (Després et al., 2012). Also, airborne fungi are common in both continental and marine environments with *Cladosporium*, *Alternaria*, *Penicillium*, *Aspergillus*, and *Epicoccum* the main identified genus (Després et al., 2012). Besides bacteria and fungal spores, viruses, algae, and pollen have also been identified in the bioaerosol of marine environments (Després et al., 2012; Fröhlich-Nowoisky et al., 2015; Michaud et al., 2018).

Therefore, the concentration, ice nucleating abilities, and variability of tropical bioaerosol need to be better characterized to quantify their role in cloud formation and precipitation development at regional level and within the tropical zonal band.

The Yucatan peninsula, surrounded by the Gulf of Mexico to the West and by the Caribbean sea to the East, with a large variety of tropical vegetation, is a great source of both terrestrial and marine

microorganisms (Guzmán, 1982; Videla et al., 2000; Morales et al., 2006). Tropical cyclones (TC) and cold fronts are some of the meteorological phenomena that seasonally affect the Yucatan peninsula every year (Whigham et al., 1991; Landsea, 2007; Knutson et al., 2010). DeLeon-Rodriguez et al. (2013) show that TC can significantly enhance the concentration of biological particles throughout the troposphere and can also efficiently transport biological particles far away from their sources.

Moreover, Mayol et al. (2017) has shown that ocean and terrestrial microorganisms can be efficiently transported long distances from their sources over the tropical and subtropical oceans.

This study presents results of the INP concentration as a function of temperature and particle size, and the concentration and composition of biological particles at a tropical coastal site (Sisal, Yucatan), to infer the potential relevance of biological particles in mixed-phase cloud formation and
precipitation development.

## 2 Methods

### 2.1 Sampling site

Ambient aerosol particles were collected between January 21 and February 02, 2017 in the coastal village of Sisal, located in the northwest corner of the Yucatan peninsula (21°09′55″N 90°01′50″W),
as shown in Fig. 1. Sisal had 1837 inhabitants in 2015 (SEDESOL, 2015), with fishing and tourism recognized as the main economical activities. The closest industry is located approximately 25 km away from the village and the nearest city is Merida, 75 km away.

The instruments used in this study were located on the roof of the Engineering Institute building of the Universidad Nacional Autonoma de Mexico (UNAM, Sisal Campus) which is 50 m from the
shoreline and about 1.7 km from the center of Sisal (Fig. 1). The roof is 25 m above ground level and directly faces the ocean.

January and February are part of the cold dry season in Mexico, with isolated events of rain associated with cold fronts reaching the deep tropics. The arithmetic mean $\pm$ standard deviation for air temperature and relative humidity (RH) during the sampling period were 22.3±3.6 °C and
68.9±6.2 %, respectively.

### 2.2 Instrumentation

A suite of instrumentation was deployed in Sisal to characterize the aerosol chemical composition, concentration, size distribution, biological content, INPs concentration, and meteorological variables (Table 1). Most instruments were run simultaneously and next to each other (less than 10 m apart)
and only wet aerosol particles were sampled (mean RH=69 %). Additionally, none of the instrumentation used an impactor or cyclone ahead of their inlets. The inlets were located around 1.5 m - 2.0 m above the roof surface. The meteorological data was obtained with a meteorological station (Davis, VANTAGE PRO2) placed in a different building approximately 20 m away from the other instruments.

### 2.2.1 Aerosol concentration and size distribution

The aerosol particle concentration and size distribution was monitored with a condensation particle counter (CPC 3010, TSI) and with an optical particle counter (LasAir II 310A, PMS), respectively. In the CPC, the size of the aerosol particles is increased in a heated saturator/cooled condenser system prior to their detection. The particles grown are directed towards a laser beam and the dispersed

light is collected by a photodetector that converts it to particle concentration. Similar to the CPC, aerosol particles in the LasAir are counted by passing them through a laser beam (without any prior treatment). Based on the pulses (or voltage) and their amplitude the dispersed light by the particles is then converted to particle concentration and size. The total particle concentration reported by the CPC was collected every second at a flow rate of 1 L min$^{-1}$, whereas the aerosol concentration as

function of their optical diameter (cut-sizes at 0.3 $\mu$m, 0.5 $\mu$m, 1.0 $\mu$m, 5.0 $\mu$m, 10.0 $\mu$m, and 25 $\mu$m) was recorded every 11 s with the LasAir at a flow rate 28.3 L min$^{-1}$.

### 2.2.2 Ice nucleating particles

Aerosol particles were collected on hydrophobic glass cover slips (HR3-215; Hampton Research) with the help of a Micro-Orifice Uniform Deposit Impactor (MOUDI 110R, MSP) to determine INP

concentrations in ambient air. Identical substrate holders as those described in Mason et al. (2015a) were used to keep the glass cover slips at a location on the impaction plate where particle concentrations varied by a relatively small amount. The MOUDI has eight stages for particle separation and collection as a function of their aerodynamic diameter (cut-sizes are 10.0 $\mu$m, 5.6 $\mu$m, 3.2 $\mu$m, 1.8 $\mu$m, 1.0 $\mu$m, 0.56 $\mu$m, 0.32 $\mu$m, and 0.18 $\mu$m). The particle size range for each MOUDI stage are

given in Table S1. The flow through the MOUDI is 30 L min$^{-1}$ and the typical sampling time was 6 h. It has been recognized that when sampling with a MOUDI under dry conditions (i.e., RH below approximately 60 %), aerosol particles can bounce from the impaction plates moving to lower stages (Winkler, 1974; Chen et al., 2011; Bateman et al., 2014). Although this is a known artifact when using this technique, this may not have been an issue in the current study given that the ambient RH

was typically above 67 %. The glass substrates containing the ambient aerosol particles were stored in petri dishes at 4 °C prior to their analysis.

The INP concentrations were measured with a cold cell coupled to an optical microscope with an EC Plan-Neofluar 5 X objective (Axiolab, Zeiss) following the MOUDI-DFT method described by Mason et al. (2015a). The cold cell-microscope system used here is the same one used in previous

studies (Mason et al., 2015a,b, 2016; DeMott et al., 2016; Si et al., 2018). The following steps encompass the analysis: i) The samples collected on glass cover slips were placed in the cold cell at room temperature, ii) The cold cell was isolated and kept at 0 °C, while humid air (RH>100 %) was injected into the cell to induce liquid droplets formation by water vapor condensation; iii) Dry air (N$_2$) was then injected into the cold cell to prevent the newly formed droplets from touching. This is

a key step to minimize the probability of liquid droplets freezing by contact; and iv) Once droplets sizes and thermodynamic conditions were stable, the cold cell was closed. The activation scans were conducted between 0 °C and -40 °C at a cooling rate of -10 °C per minute for particles collected on stages two to seven. Stage one (¿10.0 $\mu$m) was not taken into account given that the aerosol concentration on the glass substrates was typically very low, whereas in stage eight (0.18 $\mu$m - 0.32 $\mu$m)

the number concentration of particles deposited on the glass substrates was so high that inhibited the

proper formation of water drops. The temperature at which each droplet froze was determined by analyzing the video from the CCD camera (XC-ST50, Sony) connected to the microscope and the data reported by the resistance temperature detector (RTD) located at the center of the cold cell with a $\pm 0.2$ °C uncertainty (Mason et al., 2015b). Homogeneous freezing experiments were performed

on laboratory blanks exposed during the preparation of the MOUDI, while heterogeneous freezing experiments were run on ambient particles deposited on the glass cover slips (Fig. S1). The [INP] was calculated using the following expression:

$$[INPs(T)] = -ln\left(\frac{N_{\mathrm{u}}(T)}{N_{\mathrm{o}}}\right) \cdot \left(\frac{A_{\mathrm{deposit}}}{A_{\mathrm{DFT}}V}\right) \cdot N_{\mathrm{o}} \cdot f_{\mathrm{ne}} \cdot f_{\mathrm{nu,0.25-0.10\ mm}} \cdot f_{\mathrm{nu,1\ mm}}, \tag{1}$$

where $N_{\mathrm{u}}(T)$ is the number of unfrozen droplets at temperature $T$, $N_{\mathrm{o}}$ the total number of droplets,

$A_{\mathrm{deposit}}$ the total area of the aerosol deposit on the hydrophobic glass cover slip, $A_{\mathrm{DFT}}$ the area of the hydrophobic glass cover slip analyzed in the DFT experiments, $V$ the total volume of air sampled, $f_{\mathrm{ne}}$ a correction factor to account for uncertainty associated with the number of nucleation events in each experiment, $f_{\mathrm{nu,0.25-0.10\ mm}}$ and $f_{\mathrm{nu,1\ mm}}$ a non-uniformity factor which corrects for aerosol deposit inhomogeneity at a scale of 0.25 mm - 0.10 mm, and 1 mm, respectively (Mason

et al., 2015a). The upper and lower detection limits of the MOUDI-DFT are 30 L$^{-1}$ and 0.01 L$^{-1}$, respectively. We refer the readers to Mason et al. (2015a) and Mason et al. (2015b) for more details of the MOUDI-DFT operational principle.

### 2.2.3 Chemical composition

A second eight-stage MOUDI (100NR, MSP) was operated simultaneously to collect aerosol parti-

cles for chemical composition analysis with particle sizes ranging from 0.18 $\mu$m to 10.0 $\mu$m. Particles were collected on 47 mm Teflon filters (Pall Science) for 48 h at a flow rate of 30 L min$^{-1}$. Filters were weighed prior and after the sampling and stored in petri dishes at 4 °C until they were analyzed. Two different analyses were performed on each filter: elemental composition, followed by ion-cation concentration analysis.

Elemental composition of the aerosol samples was determined by X-ray fluorescence (XRF), using the x-ray spectrometer at Laboratorio de Aerosoles, Instituto de Fisica, UNAM (Espinosa et al., 2012). The samples were mounted on plastic frames with no previous treatment. The analysis was carried out using an Oxford Instrument (Scotts Valley, CA, USA) x-ray tube with an Rh anode and an Amptek (Bedford, MA, USA) Silicon Drift Detector (resolution 140 eV at 5.9 keV). The tube

operated at 50 kV and a current of 500 $\mu$A, irradiating during 900 s per spectrum. The efficiency of the detection system was measured using a set of thin film standards (MicroMatter Co., Vancouver, Canada). The spectra obtained for the samples were deconvolved with the WinQXAS computer code (IAEA, 1997), and the experimental uncertainties in elemental concentrations were computed according to the method described by Espinosa et al. (2010).

After the XRF analysis, the Teflon filters were analyzed for $NO_3^-$, $SO_4^{2-}$, $Cl^-$, $K^+$, $Na^+$, $Ca^{2+}$,

$Mg^{2+}$, and $NH_4^+$ using a Dionex model ICS-1500, equipped with an electrical conductivity detector, following Chow and Watson (1999). $NO_3^-$, $Cl^-$, and $SO_4^{2-}$ were separated using a Thermo Scientific Dionex IonPac AS23-4 $\mu$m Analytical Column (4 mm x 250 mm) with Thermo Scientific Dionex CES 300 Capillary Electrolytic Suppressor module. The injection volume was 1000 $\mu$L, mobile phase was 4.5 mM $Na_2CO_3$ - 0.8 mM NaHCO at 1 mL min$^{-1}$ flow rate. For $NH_4^+$, $Na^+$, $Ca^{2+}$, $Mg^{2+}$, and $K^+$, volumes of 1000 $\mu$L were injected in a Thermo Scientific Dionex IonPac CS12A Cation-Exchange Column (4 mm x 250 mm) with the Thermo Scientific Dionex CES 300 Capillary Electrolytic Suppressor. The mobile phase was a solution $CH_4SO_3$ 20 mM and 1 mL min$^{-1}$ flow rate.

### 2.2.4 Biological particles

Air samples were collected using two Quick Take 30 Sample Pump BioStage viable cascade impactor (SKC Inc. USA), which is a one-stage portable battery-powered instrument operated at a constant airflow rate (28.3 L min$^{-1}$) for a sampling time of 5 min. Petri dishes containing Trypticase soy agar (TSA; BD Bioxon) media, supplemented with 100 mg L$^{-1}$ cycloheximide (Sigma-Aldrich) to prevent fungal growth, were used for capture cultivable total bacteria, and Malt extract agar (EMA; BD Bioxon) for cultivable airborne propagule fungi. The two impactors, one with the TSA and the other one with EMA growing media, were run in parallel. After exposure, the plates were incubated at 37 °C during 24 h - 48 h for cultivable total bacteria and at 25 °C during 48 h - 72 h for propagule fungi. After incubation, colonies growing on each plate were counted and concentrations were calculated by taking the sampling rates into account and reported as colony-forming units per cubic meter (cfu m$^{-3}$) of air. The petri dishes with the grown colonies were stored at 4 °C prior to their analysis. Fungi were identified to genus level by macroscopic characteristics of the colonies and microscopic examination of the spore structure. Representative bacterial colonies were selected and purified using several transfer steps of single colonies on TSA and checked by Gram-staining and microscopy. Fresh biomass of the bacterial isolates were suspended in 30 % glycerol LB-broth (Alpha Biosciences, Inc.) and stored at -72 °C, for further analyses.

Bacteria isolated from the pure cultures were identified by 16S rRNA sequencing. DNA was extracted using the QIAamp DNA Mini kit (QIAGEN), according to the manufacturers protocol. Partial 16S rRNA gene sequences were amplified by polymerase chain reaction (PCR) using universal bacterial primers 27F (5-AGA GTT TGA TCM TGG CTC AG-3) and 1492R (5-TAC GGY TAC CTT GTT ACG ACT T-3) (Lane, 1991). PCRs were performed in a total volume of 50 $\mu$L including 2 $\mu$L of bacterial DNA, 35.4 $\mu$L of ddH$_2$O, 5 $\mu$L of 10 X buffer, 1.5 $\mu$L of MgCl$_2$ (1.5 mM), 1 $\mu$L of dNTPs (10 mM), 0.1 $\mu$L of Taq DNA polymerase (5 U $\mu$L$^{-1}$), and 2.5 $\mu$L each primer (10 $\mu$M). Cycle conditions were as follows: initial denaturation at 94 °C for 1 min; followed by 35 cycles at 94 °C for 1 min, 56 °C for 30 s, 72 °C for 1.5 min; and a final extension at 72 °C for 5 min. The PCR products were examined for size and yield using 1.0 % (w/v) agarose gels in TAE buffer. After

successful amplification, the obtained products were sequenced using a PRISM 3730 automated sequencer (Applied Biosystem Inc.). DNA sequences were edited and assembled using the Seq-Man and Edit Seq software (Chromas Lite, Technely Slom Pty Ltd. USA). Sequence similarity analysis was performed using the BLAST software (http://www.ncbi.nlm.nih.gov/BLAST).

Although specific growing media for actinobacteria were not used in this study, some actinobacteria colonies were able to grow on the TSA petri dishes; therefore, in some cases they were isolated and identified as follows. Genomic DNA was extracted using standard protocols reported previously for actinobacteria (Maldonado et al., 2009). The DNA preparations were then used as template for 16S rRNA gene amplification using the universal set of bacterial primers 27f and 1525r (Lane, 1991). The following components for the PCR mix were employed: 0.5 $\mu$L DNA template (for a final concentration of 100 ng $\mu$L$^{-1}$, 5 $\mu$L 10X DNA polymerase buffer, 1.5 $\mu$L MgCl$_2$ (50 mM stock solution), 1.25 $\mu$L dNTP (10 mM stock mixture), 0.5 $\mu$L of each primer (20 $\mu$M stock solution), 2.0 units of Taq polymerase made up to 50 $\mu$L with deionized sterile distilled water.

The PCR amplification was achieved using a Techne 512 gradient machine using the protocol described in Maldonado et al. (2008). The expected product (size aprox 1,500 bp) was checked by horizontal electrophoresis (70 V, 40 min) and then purified using the QIAquick PCR purification kit (QIAGEN, Germay) following the manufacturers instructions. Purified 16S rRNA gene PCR products were sent for sequencing to Macrogen (Korea) for the DyeDeoxy Terminator Cycle Sequencing kit (Applied Byosystems). Assembly of each 16S rRNA gene sequence was performed using Chromas (www.technelysium.com.au) and checked manually with the SeaView software (Galtier et al., 1996). Each assembled sequence was compared against two databases, namely, (a) the GenBank database (www.ncbi.nlm.nih.gov) by using the BLAST option and (b) the EZCloud (www.ezbiocloud.net) under its EZTaxon option. Both databases generated a list of the closest phylogenetic neighbors to each sequence and the EZTaxon, specifically provided the list of the closest described (type) species. At least 650 bp were employed for the analyzes.

## 3 Results and discussion

### 3.1 Aerosol concentration and meteorology

Two cold fronts affected Sisal during the sampling period between, January 21 and February 02, 2017, providing different air mass characteristics. The periods affected by each of the fronts are indicated in Fig. 2 by vertical grey bars and labeled Cold Front A and Cold Front B, associated with increased wind speed and shifts in wind direction. Figure 2B-D shows the time series of the aerosol particle concentration between January 21 and February 02, 2017. There is a large diurnal variability for the aerosol particle concentration measured by the CPC (particles > 30 nm, Fig. 2B) and the LasAir (particles >300 nm, Fig. 2C). Assuming log-normal distributions, the geometric mean concentration and multiplicative standard deviation (c.f. Limpert et al. (2001)) for the entire

sampling period was 758.51 $^x$/ 1.76 cm$^{-3}$ and 1.00 $^x$/ 1.37 cm$^{-3}$, respectively. From the CPC data shown in Fig. 2B, there seems to be a daily cycle with most of the highest concentration taking place between 7 h and 12 h (local time), most notably on days without the influence of cold fronts.

The data reported by the CPC and the LasAir indicate that most of the aerosol particles were smaller than 300 nm. A similar result was found by Rosinski et al. (1988) in the Gulf of Mexico (GoM) who found that the aerosol concentration for particles ranging between 0.5 $\mu$m and 1.0 $\mu$m was three to four orders of magnitude smaller than particles ranging between 0.003 $\mu$m and 0.1 $\mu$m. A decrease in aerosol particle concentration was observed at the arrival and during the passage of two cold fronts

during the sampling period, associated with an increase in horizontal wind speed of at least a factor of three (Fig. 2A). During the passage of cold front A, precipitation events were not observed which was not the case for cold front B. This could partially explain the lower aerosol concentration during the passage of the cold front B in comparison to cold front A. Also note that during the influence of the cold front A, the wind direction was almost constant from approximately 270 °, while during

cold front B the wind direction varied between 270 ° and 360 °, a more northerly component and a larger influence from the GoM compared to winds associated with cold front A.

Back-trajectories from the measurement site were estimated using the HYSPLIT model (Stein et al., 2015). They were run for 72 h for each of the days of the campaign. In the absence of cold fronts A and B, air masses arriving in Sisal had a predominantly continental influence, associated

with southerly winds (Fig. S2). However, when the cold fronts A and B reached the Yucatan Peninsula, northerly and northwesterly winds prevailed and contributed a more maritime influence. The arrival of the cold fronts was also confirmed by the surface weather maps for January 22 and 29 (Fig. S3) provided by the National Oceanic and Atmospheric Administration (NOAA).

Air masses behind both cold fronts, flowing over the GoM, were characterized by lower aerosol

particle concentrations than air masses coming from the south to the site. This result agrees well with a large body of evidence indicating that marine air masses have lower aerosol particle concentration than continentally-influenced air masses (Patterson et al., 1980; Fitzgerald, 1991). As for the total aerosol concentration (Fig. 2), the number size distributions of the aerosol particles larger than 300 nm were also impacted by the cold fronts. For example, the concentration of particles smaller

than 5.0 $\mu$m was lower during the passage of the cold front B (Fig. S4). As shown in Fig. 3 (and Fig. S5), the XRF analysis indicates that although there are small differences in the bulk chemical composition of the aerosol particles, the overall composition is generally comparable in the presence or absence of cold fronts. Note, however, that this is not a completely fair comparison given that sampling time for the chemical analysis was 48 h, while sampling time for determining the influence

of the cold front air masses on INP populations was on the order of 36 h. Therefore, the periods denoted as cold fronts contain aerosol particles that may not technically correspond to cold front air masses

## 3.2 Ice nucleating particle concentration

A total of 41 samples (8 stages each) were collected during the Sisal field campaign to calculate the [INP] as a function of temperature and particle size. Some of these samples showed a high ice nucleating activity with onset freezing temperatures found to occur at temperatures as high as -3 °C (Fig. S6). Figure 4 summarizes the [INP] as a function of temperature and particle size for 29 analyzed samples. Due to technical issues it was not possible to analyze the samples collected after January 30th. Figure 4 also shows recent literature data obtained at coastal/marine regions from DeMott et al. (2016), Welti et al. (2018), and Irish et al. (2019).

At -15 °C the [INP] measured in Sisal are in relatively good agreement with those found at Cabo Verde (Welti et al., 2018) but are one to two orders of magnitude higher than the values reported by Irish et al. (2019) from the Arctic boundary layer, and by DeMott et al. (2016) from sea spray laboratory generated particles and ambient marine boundary layer particles. As temperature decreases from -20 °C to -30 °C there is a better agreement between the Sisal [INP] and data from DeMott et al. (2016). It is important to note that the large variability of the [INP] from Welti et al. (2018) is related to the large amount of data summarized on each dotted line (i.e., from 2009 to 2013).

The "high" [INP] found at -15 °C can be explained in part by the very efficient INPs shown in Fig. S6 with sizes ranging from 1.0 $\mu$m to 1.8 $\mu$m. However, it is important to note that particles with diameters between 1.8 $\mu$m and 10 $\mu$m also contribute to the total [INP] at warm temperatures. Aerosol particles acting as INPs at -15 °C are usually biological given that other aerosol particles such as metals, crystalline salts, combustion particles (e.g., soot), and organics are not efficient INPs under these conditions. Moreover, for typical atmospheric concentrations of mineral dust, ice nucleation at these temperatures seems to be of secondary importance (Hoose and Möhler, 2012; Murray et al., 2012; Kanji et al., 2017). The potential sources of the measured INPs in Sisal are discussed below.

Figure 5 is based on Mason et al. (2016) and shows the average [INP] for three different temperatures (-15 °C, -20 °C, and -25 °C) at different locations around the globe using the same sampling and analysis methods. The Sisal data corresponds to particle diameters ranging between 0.32 $\mu$m and 10 $\mu$m; full information in all size stages was obtained in 16 out of the 29 samples analyzed. At -15 °C the average [INP] in Sisal was lower than Colby (USA), an agricultural site, and Labrador Sea; however, the obtained values are comparable to those found at UBC (Canada), Saclay (France), and Ucluelet (Canada). At -20 °C and -25 °C the average [INP] in Sisal were comparable or higher than at the other locations. As shown by the stars on top of the Sisal bars, the [INP] during the passage of the cold fronts was found to be higher than the average [INP] although the obtained values are within the uncertainty bars. For example, at -15 °C the [INP] increases from 0.33 L$^{-1}$ to 0.59 L$^{-1}$ in the cold air mass after the passage of cold front B. Recalling that the air masses behind cold front B contained a lower aerosol particle concentration, this suggests that the marine particles in that air mass are more efficient INPs than in the air masses with more continental influence. Given

that the bulk chemical composition as shown in Fig. S5 (and Fig. 3) is comparable before, during, and after the passage of the cold front B, it is possible that the observed differences in the ice nucleating abilities are linked with the biological content in the cold air masses. This is further discussed below.

The majority of the field studies performed to measure the [INP] have been conducted in midlatitudes; nevertheless, we compare here our observations with the results presented by Rosinski et al. (1988) who measured the [INP] in the condensation freezing mode for particles in the GoM during a cruise between July 20 and August 30, 1986, during mid-summer. The study reports very efficient INPs with onset freezing temperatures as high as -4 °C for particles with diameters between 0.1 $\mu$m and 0.4 $\mu$m. On August 6, 1986 (the closest sampling site to Sisal in the GoM) the study shows that the [INP] at -15 °C was on the order of $10^{-2}$ L$^{-1}$ for particles with sizes between 0.1 $\mu$m and 0.4 $\mu$m. In contrast, our results indicate that the [INP] at -15 °C varied between $10^{-1}$ L$^{-1}$ and $10^{0}$ L$^{-1}$ for particles ranging between 0.32 $\mu$m and 10 $\mu$m. This discrepancy could be attributed to the differences in the size of the particles sampled and could also be influenced by seasonal variability. If supermicron particles are excluded, the [INP] at -15 °C from the present study is one order of magnitude lower (Fig. 4). As shown by DeMott et al. (2010) particles larger than 500 nm are the more likely potential INPs and as stated by Mason et al. (2016) and as shown in Fig. 4, super-micron particles are a large contributor to the INP population. Additionally, the chemical composition of the aerosol particles collected by Rosinski et al. (1988) indicate that the air masses in the GoM in July-August were significantly influenced by mineral dust particles. African dust episodes reach Florida between May and October (Lenes et al., 2012), and there have been no reported episodes during the sampling period of this study (Jan-Feb).

### 3.2.1 [INP] vs. particle size

Figure 6 shows the mean [INP] concentration as a function of particle size between 0.32 $\mu$m and 10 $\mu$m at four different temperatures (-15 °C, -20 °C, -25 °C, and -30 °C). Note that the INP size distributions are different for each of the temperatures considered, in contrast with the results from Mason et al. (2015b) on the Pacific coast of Canada. At -15 °C the peak [INP] corresponds to particles ranging between 1.0 $\mu$m and 1.8 $\mu$m; this range has been reported as the typical size for airborne bacteria (Burrows et al., 2009). Similar size distributions were obtained at -20 °C and -25 °C with peak concentration for particles ranging in size between 3.2 $\mu$m and 5.6 $\mu$m. Finally, at -30 °C the peak was observed at smaller sizes (i.e., between 1.8 $\mu$m and 3.2 $\mu$m). The discrepancies between the present results and those from Mason et al. (2015b) at -15 °C and -30 °C could be explained by differences in airmass history. Although both studies were conducted at coastal locations, the back-trajectories from the present study indicate that during "normal" days (i.e., 70 % of the time) the sampled air masses had a significant continental contribution (Fig. S2). In contrast, air masses were mostly maritime in the Mason et al. (2015b) study. Also, it is important to note that although

the cold air masses that reached Sisal behind the cold fronts had crossed the GoM, the aerosol particles found in them are likely a mixture of particles originated in the US Central Plains and the GoM (Figs. S2B-C and S5).

Figure 6 also shows that most of the INPs are in the supermicron size range, where submicron particles represent less than 10 % of the total [INP] independent of temperature, in agreement with Mason et al. (2015b) and Mason et al. (2016). To confirm the size dependence and the importance of supermicron particles to the [INP] in Sisal, the fraction of particles acting as INPs was calculated combining the DFT and LasAir data (Fig.7). The [INP] was normalized for four size bins (i.e., 0.3 $\mu$m - 0.5 $\mu$m, 0.5 $\mu$m - 1.0 $\mu$m, 1.0 $\mu$m - 5.0 $\mu$m, and 5.0 $\mu$m - 10.0 $\mu$m). As expected (from Figs. 4 and 6), the fraction of particles acting as INPs increases with increasing particle size and with decreasing temperature. This trend is in agreement with the results shown by Si et al. (2018), with the present results being higher. Figure 7 also shows that the fraction of aerosol particles acting as INP is higher when influenced by the cold fronts (black symbols), especially for particles ranging between 1.0 $\mu$m and 5.0 $\mu$m.

### 3.3  Identification of the potential INP sources

The chemical analyzes of the sampled aerosol particles (for the whole sampling period) indicate that a large fraction of the particle mass (for sizes between 0.18 $\mu$m and 10.0 $\mu$m) are likely of marine origin (Figs. 3 and 8(A-B)). Both techniques, i.e., XRF and HPLC found that the main elements and ions are sodium and chlorine. The low concentrations of Ti, Cu, K, and Zn shows the very low probability of anthropogenic influence at the sampling site. However, although sulfate and ammonium can be emitted by natural sources, their presence, in addition to nitrates, indicate that the influence of anthropogenic activities to the aerosol population is not completely negligible. Finally, the low concentration of Al, Fe, Ca, and Si suggest that mineral dust is not a major contributor of aerosol particle mass during the sampling period. However, although the long-range transport of mineral dust particles from Africa to the Yucatan Peninsula and the GoM is very rare between January and February, mineral dust particles are frequently found in the Caribbean including the GoM (Rosinski et al., 1988; Prospero and Lamb, 2003; Doherty et al., 2008; Kishcha et al., 2014).

Figure 8(C-D) shows the mean mass size distribution for the whole sampling period of the main five elements/ions determined by the XRF and HPLC techniques. For the XRF analyzes Na, Cl, and Ca, have a single peak at 3.2 $\mu$m, whereas the S and Mg reported two peaks at 0.32 $\mu$m and 3.2 $\mu$m. Similar to the XRF results, the HPLC analyzes for Na$^+$ and Cl$^-$ also showed a single peak at 3.2 $\mu$m. SO$_4$$^{2-}$, NO$_3$$^-$ showed two peaks at 0.32 $\mu$m and 3.2 $\mu$m, whereas for NH$_4$$^+$ the peaks were located at 0.32 $\mu$m and 5.6 $\mu$m. The obtained size distributions are in agreement with those of sea salt type particles as reported elsewhere (O'Dowd et al., 2004; Prather et al., 2013).

Although Al, Si, Ca, and Fe were found in low concentrations (Fig. 8A), Tables 2 and S2 suggest that mineral dust particles are an important source of INPs in Sisal at temperatures ranging from

-20 °C to -30 °C. This is in close agreement with the results obtained by Si et al. (2019) in the Canadian high Arctic. From the correlation of the [INP] and the aerosol chemical composition at -15 °C, Mg was the only element showing a correlation that is statistically significant at the 95 % confidence interval (p<0.05). Although Mg can be found in mineral dust particles in low percentages, it can also be found in marine environments linked to sea spray aerosol (e.g., Savoie and Prospero (1980); Andreae (1982); Casillas-Ituarte et al. (2010)). Given that mineral dust particles are unlikely the source of the measured INPs above -15 °C (as suggested by Table 2), and as secondary organic aerosol and soot are not typically efficient INPs at temperatures above -15 °C (Kanji et al., 2017), in addition to the supermicron size of ca. 90 % of the INPs (Fig. 6), bioaerosol is a potential source of the INPs measured at warm temperatures. Note that bioparticles have been shown to efficiently nucleate ice at those high temperatures (Hoose and Möhler, 2012; Murray et al., 2012; Ladino et al., 2013). Efficient INPs such as those measured in Sisal could be very important for cloud glaciation. Additionally, they can trigger ice multiplication or secondary ice formation at such high temperatures via the Hallett-Mossop mechanism (Hallett and Mossop, 1974; Field et al., 2017) and impact precipitation formation.

To confirm the presence of bioparticles around Sisal and to determine their potential role in the ice nucleating abilities of the collected aerosol particles, bacteria and fungi identification was performed. As stated by Islebe et al. (2015) both bacteria and fungi need to be properly documented in the peninsula and the GoM to fully understand their regional importance. Samples for viable bacteria and fungi were collected every day at 6:00, 8:00, 10:00, and 12:00 local time. However, a single daily profile was performed between January 22 and 23. Bacteria and fungi colony forming units (cfu) m$^{-3}$ were usually above zero with the highest concentrations found early in the morning (Fig. S7). The bacteria and fungi concentrations showed a relatively good correlation between each other (r=0.55, p<0.0005 not shown) with average values for the whole sampling period of 295±312 cfu m$^{-3}$ and 438±346 cfu m$^{-3}$, respectively. The bacteria concentration are comparable to the values found by Hurtado et al. (2014) in Tijuana, on the Pacific coast of Mexico (i.e, 230 cfu m$^{-3}$ - 280 cfu m$^{-3}$). Bacteria and fungi concentrations were found to be lower when the wind was coming from the north in comparison with southern-continental air masses (Fig. S8), a behavior similar to the aerosol concentration shown in section 3.2.

Figure 9 shows the time series of the [INP] together with the bacteria and fungal concentrations. Panels B and C show a poor correlation between the bacteria and fungal concentrations with the [INP] with correlation coefficients at -15 °C of 0.12 (p=0.06) and 0.36 (p=0.03), respectively. This poor correlation can be in part due to the different sampling time of the MOUDI and the biosamplers. Another additional factor is the fact that the reported bacteria and fungi concentrations are only a small fraction of the total population given that the used method is selective to viable microorganisms only. Note that the fraction of detected microorganisms by culture methods are typically ca. 1 % (but can be lower) of the total population (Lighthart, 2000; Burrows et al., 2009). From Fig. 9 it is

notable that although the bacteria and fungi concentrations were very low on January 29 (i.e., under the influence of the cold front B), the [INP] at -15 °C was comparable to the average value for the entire campaign. It is therefore intriguing if the marine microorganisms brought to Sisal by the cold front B could be efficient INPs.

Table 3 summarizes the identified bacteria before the arrival of cold front A and after the passage of cold fronts A and B. Additionally, Table 4 shows the fungi identification for the whole campaign. To our knowledge this is the first time that airborne viable bacteria, and fungi are identified at this coastal location. Although biological microorganism characterization has been previously conducted in Mexico, those studies focused on health effects mainly (Santos-Burgoa et al., 1994; Guzman, 1998; Maldonado et al., 2009; Frías-De León et al., 2016; Ríos et al., 2016). Note that 76 % of the detected bacteria were Gram positive with *Micrococcus*, *Staphylococcus*, and *Bacillus* as the main identified genus (Fig. S9). As shown in Table 3, before the arrival of cold front A (January 21-22), a large variety of bacteria species were found with different typical sources, mostly terrestrial. This is in contrast with the identified species found after the passage of cold fronts A and B. Especially, after cold front B different *Vibrio* species were identified, most of which are typically of marine origin. Recently, Hurtado et al. (2014) found that the most common genera of the bacteria in Tijuana were *Staphylococcus*, *Streptococcus*, *Pseudomonas*, and *Bacillus* in close agreement with the present results.

Regarding fungi, different genus were also identified as shown in Table 4 with *Cladosporium* and *Penicillium* as the most frequent ones (51 % and 11 %, respectively) as shown in Fig. S9. This is in good agreement with the data reported by Després et al. (2012).

Several studies haven shown the good correlation between the concentration of fluorescent biological particles and the [INP]; however, from those studies it is highly uncertain if the good ice nucleating abilities can be attributed to a single microorganism specie (Mason et al., 2015b; Twohy et al., 2016). Off-line methods as the one used here have been able to identify from rain water and cloud water specific microorganisms such as *Pseudomonas syringae*, *Micrococcus*, *Staphylococcus*, *Cladosporium*, *Penicillium*, *Aspergillus* among others, with some showing good ice nucleating ability (Amato et al., 2007, 2017; Delort et al., 2010; Failor et al., 2017; Stopelli et al., 2017; Akila et al., 2018).

## 4 Conclusions

Aerosol particles collected around Sisal (on the northwest coast of the Yucatan peninsula) from January 21 to February 02, 2017 were found to be efficient INPs with onset freezing temperatures as high as -3 °C, similar to the onset freezing temperature of the well known efficient INP *Pseudomonas syringae* (Wex et al., 2015) and Arctic sea surface microlayer organic-enriched waters (Wilson et al., 2015). The results show that the INP concentrations in Sisal are comparable (geometric mean and

multiplicative standard deviation of 0.44 $^x$/ 1.77 L$^{-1}$, 1.73 $^x$/ 2.56 L$^{-1}$, and 6.20 $^x$/ 2.65 L$^{-1}$ at -15 °C, -20 °C, and -25 °C, respectively) and in specific cases even higher than at other locations studied using the same type of INP counter. Higher INP concentrations were observed especially under the influence of cold fronts. This is an intriguing result given that the air masses behind the cold front contained lower aerosol particle concentrations. This deserves further analysis given that the Yucatan peninsula and the Caribbean region are impacted regularly by this meteorological phenomenon during the winter and early spring months.

The chemical analyzes performed on the sampled aerosol particles did not indicate the presence of mineral dust particles in high concentrations (the combined mass concentrations of Al, Si, and Fe correspond to 5.1 % of the total particle mass measured by the XRF). However, Al, Si, Ca, and Fe showed high correlation coefficients ($r^2$ above 0.64 with p<0.05) with the [INP] at temperatures between -20 °C and -30 °C. At -15 °C the [INP] in Sisal was one to two orders of magnitude higher than the concentrations reported from other coastal/marine regions around the globe. The size of this very efficient INPs was found to be above 1.0 $\mu$m with a large contribution to the [INP] of particles ranging from 1.0 $\mu$m to 1.8 $\mu$m. Based on the large [INPs] above -15 °C, the supermicron size of 90 % of the INPs, the presence of marine biological particles in the cold air masses those of which showed the highest [INP], in addition to the poor correlation shown by the mineral dust tracers with the [INP], we hypothesize that the likely source of the INPs measured in Sisal at high temperatures are biological particles. Therefore, our results suggest that continental and maritime biological particles could play an important role in ice cloud formation and precipitation development in the Yucatan peninsula. Although several bacteria and fungi were identified, it is unknown if any of them were responsible for the observed ice nucleating abilities of the aerosol around Sisal.

The present results are important for the development of new parametrizations to be incorporated in climate models given that the currently available parametrizations contain little or no data from tropical latitudes. However, further similar studies are needed given that the [INP] may vary seasonally. Especially, the arrival of mineral dust particles to the GoM and the Caribbean region from Africa in July-August are expected to impact the [INP] and therefore, ice cloud formation, as shown by Rosinski et al. (1988) and DeMott et al. (2003).

The quantitative understanding of the importance of biological particles in ice particle formation is a challenging task for the cloud physics community. As shown here, even when combining biology with chemistry, physics, and meteorology, the results obtained are not as quantitative as would be desired. Therefore, further studies are needed in order to improve our current limited understanding of the role that tropical microorganisms could play in ice cloud formation.

*Data availability*. Data are available upon request to the corresponding author.

*Author contributions*. LAL and GBR designed the experiments. LAL, HAO, MAE, and BF carried out the INP and aerosol measurements. IR, LM, ES,EQ, LAM, and AGR analyzed the biological particles. HAO and JM performed the chemical analyzes. LAL, ZRD, CC, AKB, MS and

VI performed the INP analyzes. LAL wrote the paper, with contributions from all co-authors.

*Acknowledgements.* The authors thank E. Garcia, G. Chavez, I. Gavilan, R. Gutierrez, J. Munoz, L. Landeros, F. Cordoba, W. Gutierrez, M. Garcia, M. Robles, A. Rodriguez, J.C. Pineda, L. Gonzalez, A. Prieto, T. Castro, M.I. Saavedra, and A. Cruz for their invaluable help. We also thank David S. Valdes from CINVESTAV Merida for sharing the meteorological data. Finally, we thank the National Oceanic and Atmospheric Administration (NOAA) for facilitating the use of the surface maps and the Hysplit. This study was financially supported by the Direccion General de Asuntos del Personal Academico (DGAPA) and by the Consejo Nacional de Ciencia y Tecnologia (Conacyt) through grants PAPIIT IA108417 and IN102818 and I000/781/2106.

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

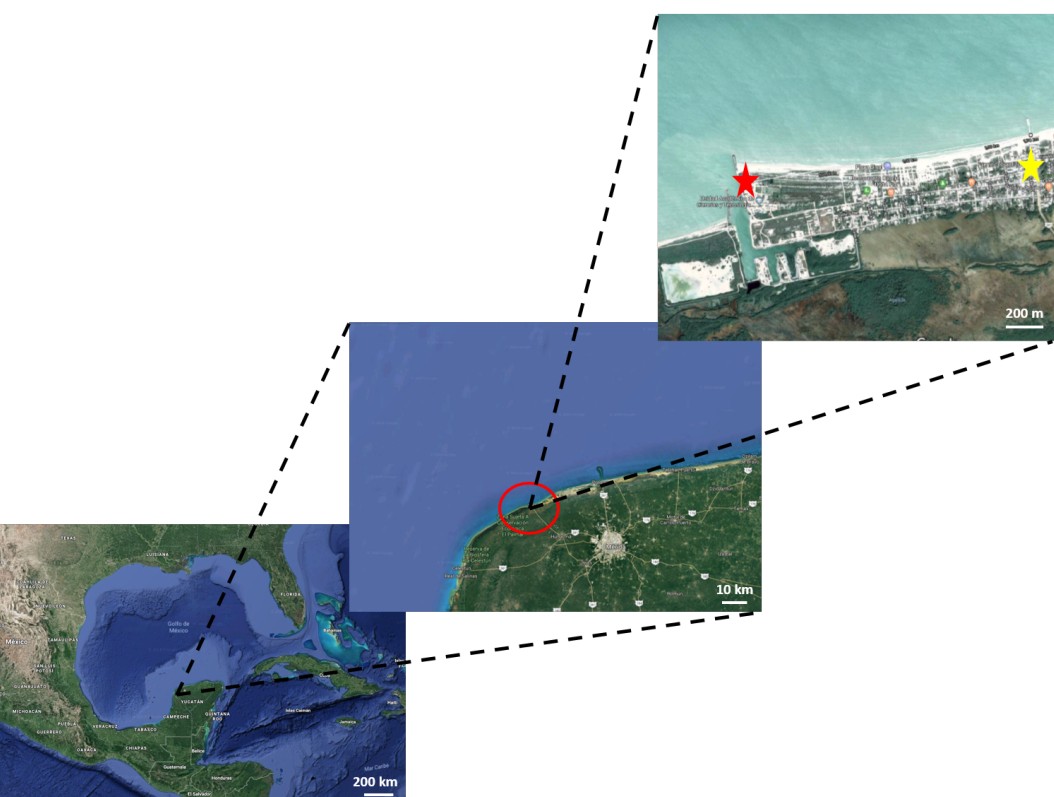

**Fig. 1.** Map showing the sampling location. The red star shows the location of the Engineering Institute building where the sampling took place, while the yellow star shows the center of Sisal (GoogleMaps).

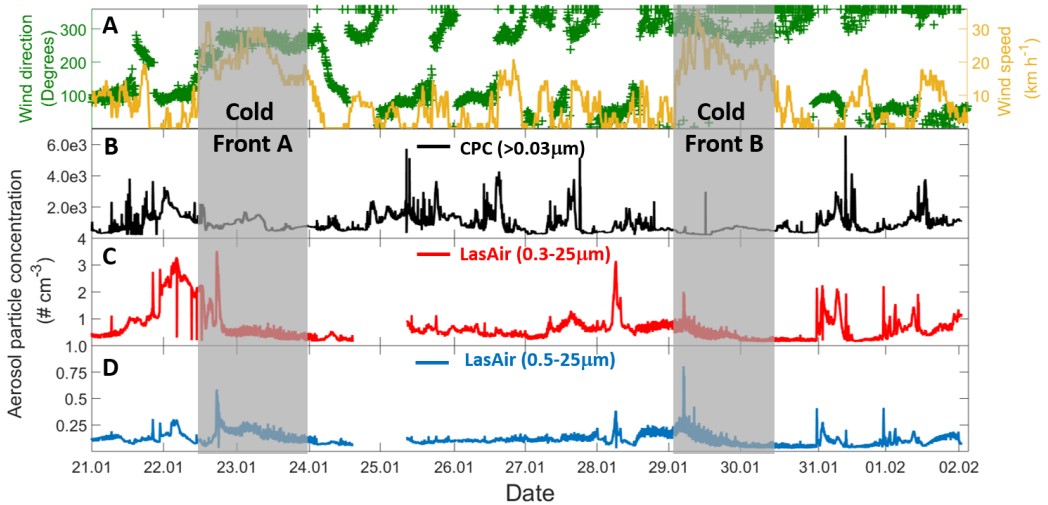

**Fig. 2.** Time evolution of wind and aerosol particle concentration time series for the entire campaign (21 January - 02 February, 2017) . A) Time series of the wind speed (yellow) and wind direction (green), B) Particle concentration measured by the CPC, C) Particle concentration measured by the LasAir full size range (0.3 $\mu$m to 25 $\mu$m), and D) Particle concentration measured by the LasAir for particles >500 nm (0.5 $\mu$m to 25 $\mu$m). Grey areas denote the periods affected by cold fronts (A and B). Each tick mark on the x-axis corresponds to midnight local time.

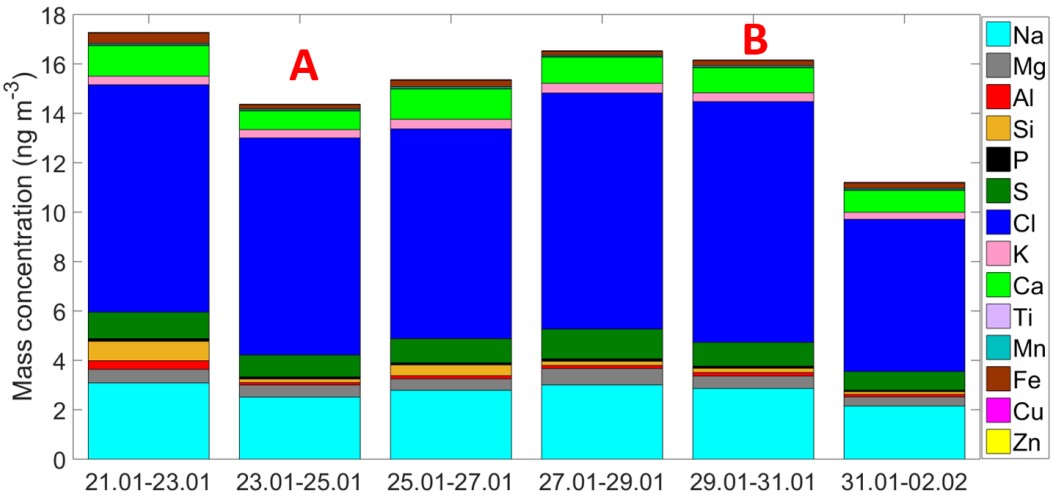

**Fig. 3.** Time series of the ambient aerosol mass concentration and bulk chemical composition as measured by the XRF. Each sample was collected for 48 h starting at 12:00 h local time. A and B indicate that those samples were partially influenced by the passage of the Cold front A and the Cold front B, respectively.

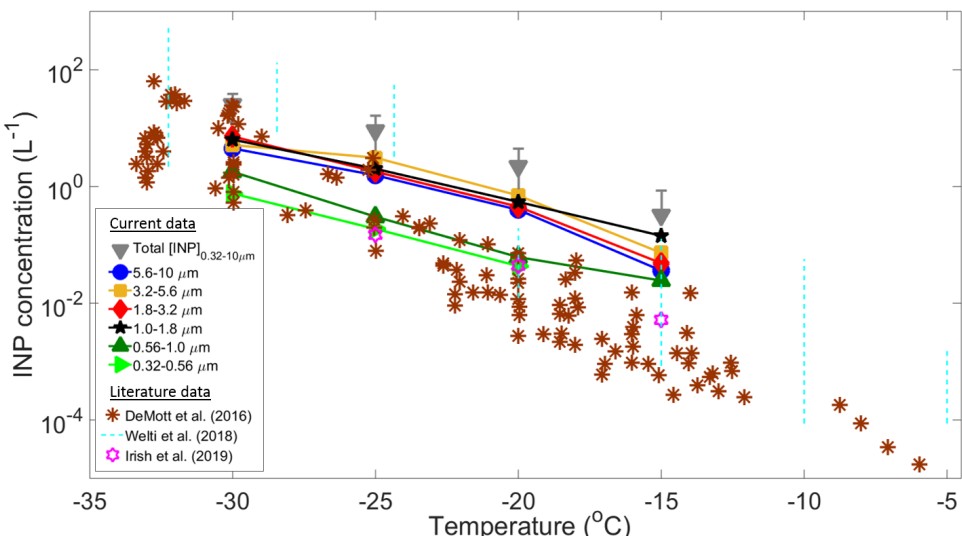

**Fig. 4.** Summary of average INP concentrations as a function of temperature and particle size (solid symbols). Total [INP] are represented by the grey triangles, whereas the brown asterisks, light blue dotted lines, and purple stars are data from DeMott et al. (2016), Welti et al. (2018), and Irish et al. (2019), respectively. The upper and lower detection limits of the MOUDI-DFT are 30 L$^{-1}$ and 0.01 L$^{-1}$, respectively.

**Table 1.** List of the measured variables and the corresponding instrumentation.

| Measured Variable | Instrument |
|---|---|
| INP concentration | MOUDI-DFT (Mason et al., 2015a) |
| Aerosol concentration | Condensation particle counter (CPC, TSI 3010) |
| Coarse aerosol size distribution | LasAir Optical particle counter (MSP) |
| Chemical composition | X-Ray fluorescence (XRF) and High-performance liquid chromatography (HPLC) |
| Bacterial and fungal concentration | Biostage impactor (SKC) |
| Meteorology | Weather station (Davis) |

**Table 2.** Correlation coefficients ($r^2$) of the average chemical composition and the average [INP] per sample at -15 °C, -20 °C, -25 °C, and -30 °C. Bold text highlights the $r^2$ with $p<0.05$ (Table S1) for each temperature. The correlations were obtained for five sample points at each temperature.

| Temperature | Na | Mg | Al | Si | P | S | Cl | K | Ca | Ti | Mn | Fe | Cu | Zn |
|---|---|---|---|---|---|---|---|---|---|---|---|---|---|---|
| -15 °C | 0.02 | **0.70** | 0.08 | 0.15 | 0.01 | 0.02 | 0.07 | 0.02 | 0.08 | 0.21 | 0.61 | 0.31 | 0.03 | 0.25 |
| -20 °C | 0.18 | 0.02 | 0.44 | **0.86** | 0.24 | 0.33 | 0.27 | **0.77** | **0.64** | 0.19 | 0.47 | **0.74** | 0.35 | 0.26 |
| -25 °C | 0.54 | 0.00 | 0.33 | 0.45 | 0.27 | 0.45 | 0.02 | 0.40 | **0.89** | 0.06 | 0.40 | 0.49 | 0.17 | 0.07 |
| -30 °C | 0.53 | 0.00 | **0.65** | **0.74** | 0.46 | 0.56 | 0.00 | **0.65** | **0.79** | 0.02 | 0.34 | **0.81** | 0.06 | 0.03 |

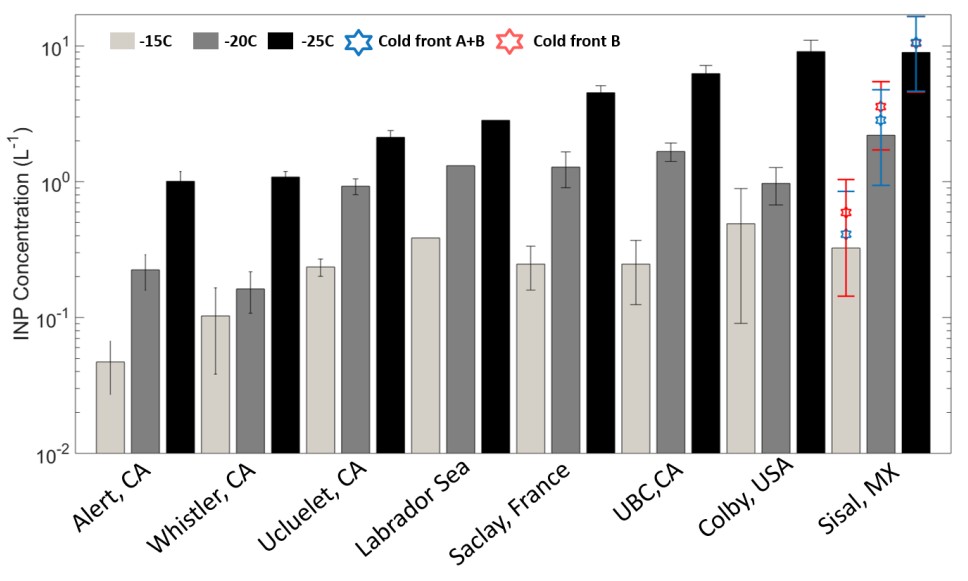

**Fig. 5.** Mean INP number concentrations at droplet freezing temperatures of -15 °C (light gray), -20 °C (dark gray), and -25 °C (black). The blue and red stars represent the mean INP concentration during the cold fronts A+B and cold front B, respectively. Uncertainties are given as the standard uncertainty of the mean (adapted from Mason et al. (2016)).

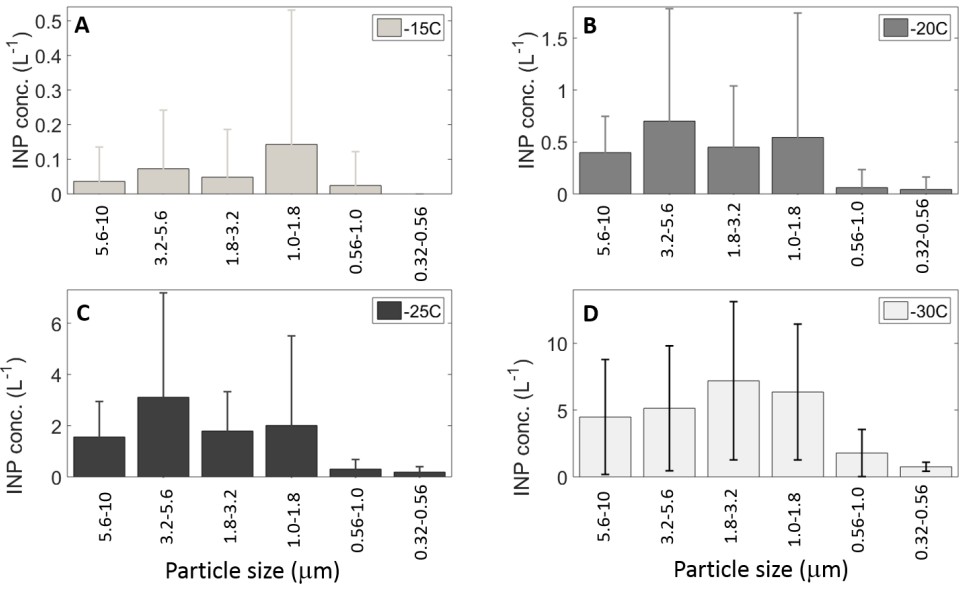

**Fig. 6.** Mean INP concentration as a function of aerosol particle size at A) -15 °C, B) -20 °C, C) -25 °C, and D) -30 °C. Uncertainties are given as the standard uncertainty of the mean.

**Table 3.** Bacterial isolation for top) Jan 21-22, middle) Cold front A, bottom) Cold front B. [a], Isolated on TSA media; [b], Isolated on GYM media.

| Phylum | Genus/species | Source |
|---|---|---|
| Actinobacteria | [a]*Kocuria palustris* | Soil, rhizoplane |
| | [a]*Micrococcus* spp | Water, soil, dust, and skin |
| | [a]*Rhodococcus corynebacteroides* | Soil, water and eukaryotic cells |
| Firmicutes | [a]*Staphylococcus kloosii* | Human and animal skin |
| | [a]*Staphylococcus lugdunensis* | Human and animal skin |
| | [a]*Staphylococcus nepalensis* | Mucocutaneous zones of humans and animals |
| | [a]*Staphylococcus arlettae* | Animal skin, mucosal zones, polluted water |
| | [a]*Staphylococcus epidermidis* | Human skin, mucosal microbiota |
| | [a]*Bacillus aryabhattai* | Upper atmosphere, rhizosphere |
| | [a]*Bacillus gibsonii* | Alkaline soil |
| | [a]*Bacillus aeris* | Soil |
| | [a]*Staphylococcus lentus* | Soil |
| Alphaproteobacteria | [a]*Sphingomonas mucosissima* | Water and soil |
| Actinobacteria | [a]*Micrococcus* spp | Water, soil, dust, and skin |
| Firmicutes | [a]*Bacillus oceanisediminis* | Marine sediments |
| Gammaproteobacteria | [a]*Proteus mirabilis* | Water and soil |
| | [a]*Pseudomonas stutzeri* | Soil |
| Actinobacteria | [a,b]*Micrococcus* spp | Water, soil, dust, and skin |
| | [a]*Micrococcus lentus* | Soil, dust, water and air |
| | [b]*Micrococcus yunnanensis* | Roots of Polyspora axillaris |
| | [b]*Streptomyces* spp | Cosmopolitan |
| Firmicutes | [a]*Bacillus* spp | Cosmopolitan |
| | [a]*Bacillus niacini* | Soil |
| | [a]*Bacillus subtilis* | Soil, gut commensal in ruminants and humans |
| | [a]*Planomicrobium koreense* | Fermented seafood |
| | [a]*Staphylococcus* spp | Human and animal skin, mucous zones, soils |
| | [b]*Solibacillus isronensis* | Air |
| | [a]*Staphylococcus equorum* | Human and animal skin |
| Gammaproteobacteria | [a]*Pseudomonas reactants* | Soil |
| | [a]*Vibrio alginolyticus* | Marine |
| | [a]*Vibrio natriegens* | Marine |
| | [a]*Vibrio neocaledonicus* | Marine |
| | [a]*Vibrio parahaemolyticus* | Marine |
| | [a]*Zobellella* sp. | Marine and estuarine environments |

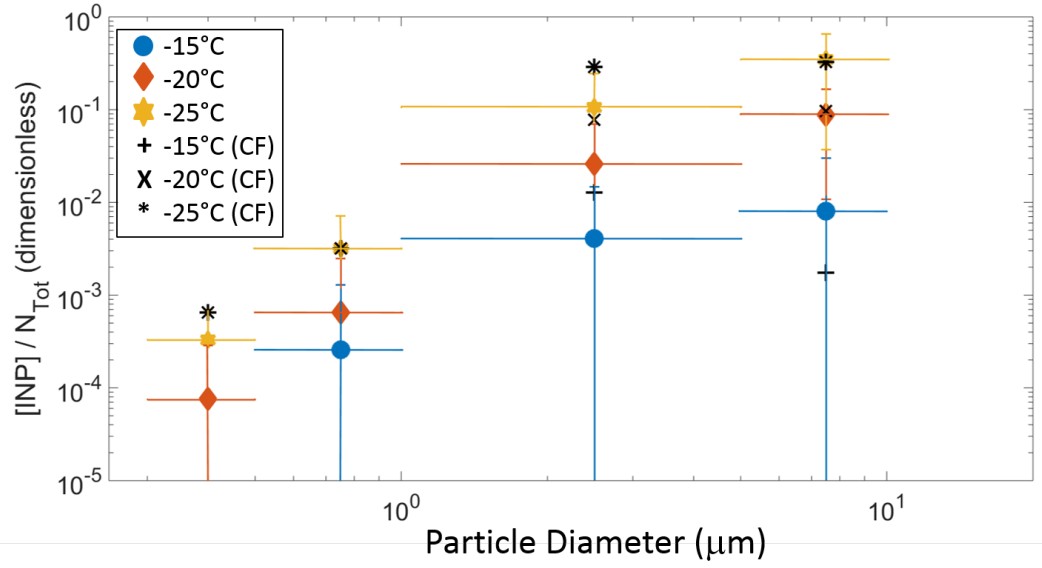

**Fig. 7.** Fraction of aerosol particles acting as an INP ([INP] / $N_{Tot}$) as a function of particle size at -15 °C, -20 °C, and -25 °C, respectively. $N_{Tot}$ refers to the number of aerosol particles in a given size range measured by the LasAir. The solid colored symbols represent the entire campaign, while the black symbols represent the samples collected under the influence of the cold fronts (CF).

**Table 4.** Fungal identification on EMA media for the whole sampling period

| Phylum | Genus | Source |
|---|---|---|
| Dothideomycetes | *Alternaria* | |
| | *Cladoporium* | |
| | *Drechslera* | Dead plants, soil, foods, air, indoor |
| Euascomycetes | *Curvularia* | enviroments, decaying organic matter, |
| Eurotiomycetes | *Aspergillus* | indoor bioaerosols, on animal systems and |
| | *Penicillium* | in freshwater and marine habitats. |
| Leotiomycetes | *Monilia* | |
| Sordariomycetes | *Fusarium* | |
| Zygomycetes | *Rhizopus* | |

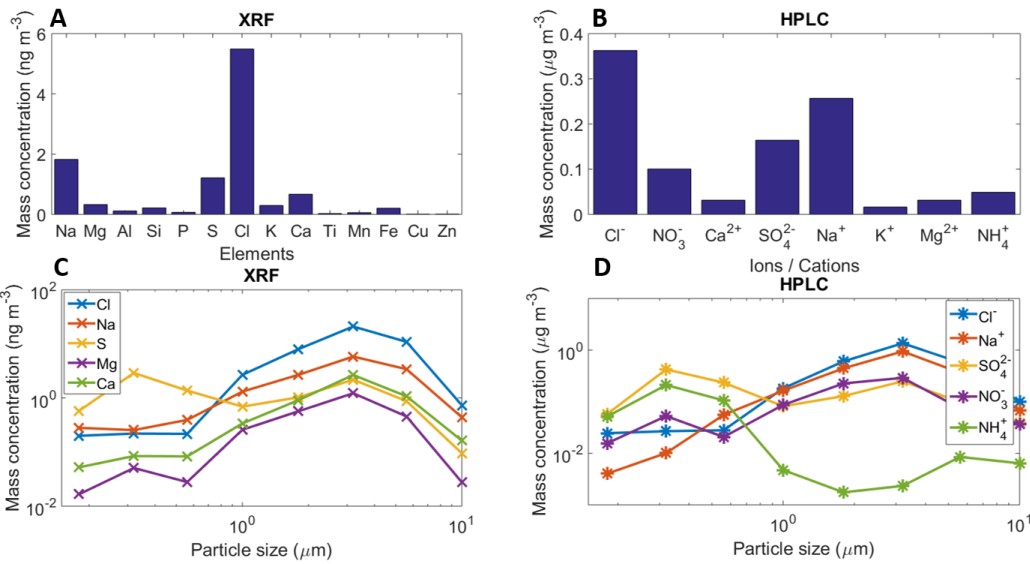

**Fig. 8.** A) Mean mass concentration of 14 detected elements for the collected aerosol particles using XRF, B) Mean mass concentration of 8 detected ions for the collected aerosol particles using HPLC, C) and D) Mean mass size distribution of the main five detected elements/ions with the XRF and HPLC, respectively. This results are the average for the whole sampling period.

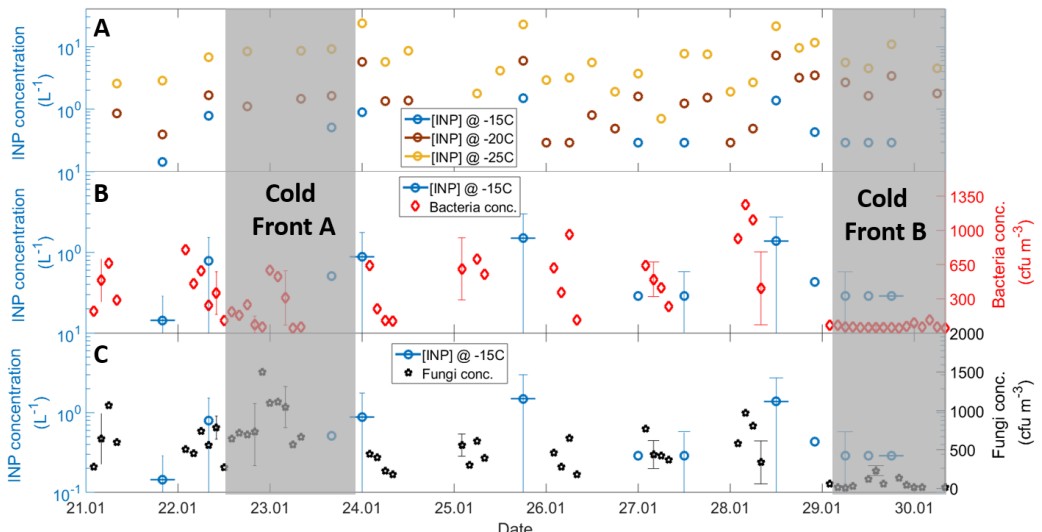

**Fig. 9.** A) Time series of the [INP] at -15 °C (blue), -20 °C (brown), and -25 °C (yellow), B) Time series of the [INP] at -15 °C (blue) together with bacteria concentration (red), and C) Time series of the [INP] at -15 °C (blue) together with fungal concentration (black). Each X-axis tick corresponds to 06:00 local time. The horizontal uncertainty bars indicate the time span the MOUDI-DFT measurements i.e., 6 h. Grey areas denote the periods affected by the entrance of a cold front (A and B).