# Peer review of "Ice Nucleating Particles in a Coastal Tropical Site"

_Atmospheric Chemistry and Physics, 2018_

## Referee Comment (RC1) · Anonymous Referee #1 · 21 Dec 2018

General comments

This manuscript presents data on physical, chemical and biological aerosol parameters observed during a campaign lasting two weeks at a coastal site on the Yucatan peninsula. Technically, the measurements were well done and there are little similar data from the same region. In general, the manuscript is clearly written.

The data obtained with the various instruments employed are weakly related, in part because of a mismatch in sampling duration and timing between INP and other parameters (chemical and biological). The sampling duration was per sample 6 hours for INP (2-3 samples a day, morning and afternoon), 48 hours for chemical components,

5 min for bacteria and fungi (4 samples a day, in the morning). Further, no link can be made between the cultured microorganisms identified (Tab. 2) and ice nucleation activity, because the cultured organisms were not tested for ice nucleation activity. Much of the relations discussed between INP and other parameters are speculation supported through reference to other literature, without the current study adding substantial new evidence in support of it. Because of that, I would like to encourage the authors to put more effort into relating the different parameters in a way that each parameter can tell us more than its individual story. To start, you could try to combine data from the optical particle counter with size resolved INP concentrations to tell for different size classes the fraction of particles that are ice nucleation active (e.g. what was the ratio INP/aerosol particle in the different MOUDI size classes? How did the ratio change during the passage of the cold front?).

Specific comments

Page 5, line 136-138: Measurements of INP concentrations with the cold cell need to be described in more detail. At least their principle should be clear to reader without having to look up the paper by Mason et al. (2015a).

Fig. 3 duplicates the time series measured by the CPC, which is already shown in Fig. 2. Combine Fig. 2 and Fig. 3, and make the time series of wind speed and direction (now panel B in Fig. 3) the top panel of the combined Figure because wind is the factor driving the aerosol concentrations, so logically this factor should come first.

Figure 4: I would like to see the data of the present study as points, not just as a shaded area, where I can not see by how many points a particular part of the area defined. In particular, I am curious to see how many points support the very high [INP] at temperatures above -10 °C. I suggest to revise the Figure in a way that the data taken from Kanji et al (2017) are shown as shaded areas only (no points) and the data of the present study are superimposed on this background as points (perhaps use different symbols for data obtained during the passage of cold fronts).

Page 8, first line (and page 11, line 344): The aerosol number concentrations are reported as mean plus-minus one standard deviation, assuming a normal distribution of values. Although this is common practice, it is not correct because aerosol number concentrations have a log-normal distribution. I strongly encourage the authors to apply the less common, but correct metrics (median and multiplicative standard deviation) as explained in Limpert et al. (2001; BioScience, 51, 341-352, freely available at: https://stat.ethz.ch/∼stahel/lognormal/bioscience.pdf). Why perpetuate a common mistake?

The Conclusion section is mostly a summary of the Results and Discussion section. It should go further than that.

Technical corrections

Title: Perhaps replace "importance" with "contribution"

Page 1, last line: The statement "Biological particles were likely found to be very important. . ." does not make sense to me. Do you mean "Biological particles could potentially be very important. . ."

Page 2, line 37: "Given the potential INP role of a variety of aerosol particles . . ." do you mean "Given the potential role of a variety of aerosol particles as INPs. . ."

Page 3, line 88: "presents", not "present"

Page 3, last line: Why "importance" and not simply "potential relevance"? The word "importance" is a premature value judgement here, at the end of the introduction section.

Page 5, line 128: I would turn the order of the cut-sizes the other way round, so that it follows the logic of the instrument (i.e. 10 um, 5.6 um, . . .0.18 um).

Page 6, line162: delete "with"; line 165: replace "0" (zero) in "Na2C03" with "O" (capital "o").

Page 8, line 260: change to "At temperatures..." (plural).

Page 9, line 273: "Saclay", not "Saclary".

Page 10, line 311: change "elements/cations/ions" to "elements and ions are sodium and chlorine, respectively chloride"; line 322: why "elements/cations/ions" and not just "elements and ions", cations are ions.

Page 11, line 353: "temporal mismatch of the data", not "uncertainty in the analysis".

―――――――――――――――――――――――

---

## Referee Comment (RC2) · Anonymous Referee #3 · 11 Jan 2019

General Comments: This article discusses INP measurements carried out at Sasal on the coast of the Gulf of Mexico and attempts to identify the sources of INP. They have identified the biological particles from the tropical ocean as the source of INP at measurements site when the wind direction is from the GoM. Considering the importance of the INPs and lack of measurements available over the in tropical latitudes, this article makes a compelling case for the publication. The article is well written although interpretation of measurement can be better. Agreeing with the comment by an anonymous referee, who has commented on the article exhaustively, the authors need to explain the method of INP measurement in brief since that is the main thrust of the article. The lack of correlation between size bin and INP at different temperatures in

[Figure]

Fig. 6 warrants detail analysis and in-depth discussion. I would encourage the author to normalize the INP concentration with the total number of particles in respective size bins. Similarly, the chemistry can be analyzed for bins to segregate the contribution of particles of different chemical composition.

Technical comments

Line 10: should it be tropical instead of topical(?) Line 35: It is encouraged that the author cites the original literature along with the newer cross-references. e.g. Kanji et al. (2017) cite other older references. Line 41: Using the word "most important" may not be a good idea. Line 110: It would be nice if "wet" is quantified in terms of relative humidity if the measurement is available. Line 115: Figure 2 shows three types of time series for 3 different cut off of the particles sizes. Since the instruments used employ different principles of measurements, it would be appropriate to explain the instrument principle in brief. Line 235: It should be Fig. 3 instead of Figure 3.
* * *

---

## Referee Comment (RC3) · Anonymous Referee #4 · 15 Jan 2019

This manuscript presents the properties of ambient particles and INPs in the tropical latitudes using several instruments and sampling techniques, in order to provide a comprehensive analysis of the chemistry and biological properties of the INPs. Characterizing INP concentration around the world and understanding the contribution of bioaerosols are important, and therefore I find this study a valuable contribution, especially since there is little data from tropical latitudes.

The introduction part is good and the manuscript is well written. However, at some parts of the manuscript I felt the need for more details, as in the ice nucleation part (see below). Also, I do not think that the title reflects the findings well. I think it confuses

the reader to think there is some new evidence on biological INPs, however, it was not proved that the bioaerosols are responsible for the ice nucleation activity that was detected, and I personally still not convinced. Thus, I would recommend changing the title to a more suitable one, or provide the proof for the importance of the biological particles to the INP concentrations.

I think that the parts related to the freezing experiments and results, should be revised and clarified. There is not enough technical details about the method, such as the instrumentation or the temperature uncertainty, and what are the limitation of the analysis that was done. Also - there are no error bars (or any error analysis) throughout the manuscript, which is very surprising, and must be done before resubmission.

Specific comments: Line 10: I guess should be "tropical" and not topical Line 13: for -> from Line 29: is -> was Line 40: I suggest to detail how the concentration affect on water drop formation Line 88: present -> presents Line 110: What is wet? Line 235: if possible, please explain here how was the cold front determined, and send to the supplementary figures Line 250: Please explain why do you believe that there is no significant difference and that during front did not changed much. To me it seems that their slope is different and that the concentrations are also differ. Line 265: I suggest rephrasing this sentence; especially the word important, it does not fit here. Line 332: I suggest removing or rephrasing this paragraph; it is a speculation that is not accurate. Line 353: Can you detail what is the "uncertainties of analysis" Line 451: fix reference of Chen et al. 2011. Fig.2 – I suggest to write "entire" instead of "whole" Fig 2. Is seems to me that there is daily cycle of the aerosols? For example seen clearly in days 27-28.1.17. Figure S1. I would consider replacing the use of brackets in this caption. It was not clear to me at first what I see there. Figure S3. Why there are no error bars? That way it will be easier to understand if the two distribution are actually similar.

---

## Referee Comment (RC4) · Anonymous Referee #5 · 31 Jan 2019

General Comments:

"The Importance of Biological Particles to the Ice Nucleating Particle Concentrations in a Coastal tropical Site" by Ladino et al. describes efforts to characterize the INP population and biological particles at a tropical site. These data are valuable to the community due to a lack of data in such environments. My major comment is that more information on the INP measurement detection limits, blanks, and uncertainties is needed because the paper heavily relies on these data. I have several other concerns in the interpretation of results, described below. In general, the authors motivate the study by describing the need to characterize marine INP sources in the tropics ("Very

few studies to sample INPs have been carried out in tropical latitudes, and there is a need to evaluate their availability to understand the potential role that marine aerosol may play in the hydrological cycle of tropical regions"), but I am not convinced the method deployed can measure [INP] for a remote region and I think the concentrations reported could not possibly be explained by marine aerosol. Overall, I think there are a few things that need to be clearly stated and supported consistently throughout: 1) What were the [INP] and their variability (and their detection limits)? 2) What size range corresponded to the highest [INP]? 3) what meteorological conditions, air mass histories (back trajectory and PCR results) corresponded to the highest [INP]? 4) What is the hypothesized origin of these very high [INP] and biological particles? If the authors can build up the results discussions around some clear points, I think it will be easier to follow along.

Comments:

Abstract:

Should mention the freezing mode and INP temperature range measured during this study in abstract.

L18 – I think a better way to say this is similar to how it was stated in the results section, something like "The high concentrations of INPs at warmer ice nucleation temperatures (T >-15C) and the supermicron size of the INPs suggest that biological particles may have been a significant contributor to the INP population in Sisal during this study".

Introduction:

L55: The modeling studies listed determined specific regions where oceanic sources dominated the INP population due to an absence of other types, like mineral dust.

L58: Bigg, 1973 was the first to report such a study (his data are used in Schnell and Vali, 1975).

L67 – the range may also be from different species, right?

Methods:

L142 – please provide information on the measurement detection limit and how measurement uncertainty was determined. Where there blanks collected and how were these accounted for?

Results and discussion:

Fig S3- Why are there no particles larger than 1 micron at a coastal site? Is this consistent with other studies? Are the y axis units correct?

L250 – Did you compare particle composition between the cold front/marine air mass periods and the other periods? These back trajectories shown in Figure S1 suggest that the air masses actually originated from the US Central Plains. So, you would expect a mixture of aerosol composition I think.

Fig 4 – It would be best to show the data from this work as points versus a shaded region so that the variability in [INP] is fully illustrated/reported. Are these samples background corrected? Are the [INP] for all the stages combined or each individual stage? For temperatures lower than ∼ -25C, the "bluish" region flat-lines at about 30 L-1 – is that the upper detection limit of the INP measurement? Same with lower detection limit. This figure suggests to me that the range of detection of this method is from 0.1 L-1 to 30 L-1 . Is it possible for this method to observe the concentrations reported for remote marine environments (dark blue shade, DeMott et al. (2016))? These detection limits should be noted in the methods and in the figure caption.

L260 –Should also note that the [INP] reported here are up to 3-4 orders of magnitude higher than [INP] reported for marine boundary layer measurements reported by DeMott et al., 2016.

L270 – "The Sisal data corresponds to particle diameters ranging between 0.32 $\mu$m and 10 $\mu$m where 16 out of the 29 samples fulfilled the size criteria." – please clarify what is meant by this?

L283 – what time of year where Rosinski's measurements made?

Fig 6 – Should there be standard deviation bars on these? Also, if one were to use Figure 6 and Figure S3 to determine a number fraction (which should be done as an analysis), the number fractions are bogus. Are the units of Figure S3 correct (maybe they should be per cubic centimeter)? How do you have higher [INP] than total particle counts in the same size bin?

Fig 7/L309 – Are these results for the entire study or a specific period? Please add this detail to figure caption and text. If the entire study, why not look at individual events? They were 48 hours sample, so perhaps show a timeline? Is it not possible to look at carbon or oxygen with this method?

L328 –show the timeline in the supplemental to support this statement?

Figure S4 – "Daily profile" or is this the average of two days of data (i.e., two points averaged for each time bin?

Fig 8 – why do only some [INP] points have horizontal lines? The y axis on the top two panels have errors for the lower limit label. What are the measurement uncertainties for [INP] and bacteria/fungi?

L365 – were offshore chlorophyll a concentrations elevated during this study?

L366 – I think it's great to show the utility of this method for showing the air mass history (i.e., terrestrial versus marine). I suggest pulling this forward in the introduction, as this a unique approach to identify air mass origin (e.g., In this study, we use PCR to confirm air mass history and its influence on [INP]) and also reference any other papers that have attempted this (if applicable).

Conclusions:

L387 – The dates of the study should be specified here (for those who read only the conclusions..)

[Figure]

L390 – Should report the range of [INP] for a given temperature

L390 – also similar to [INP] measured from U.S. Central Plains (harvesting aerosol), as you mention in the text. I think it should also be clear that the [INP] are high for a marine environment (i.e., comparison to DeMott et al., 2016 marine measurements).

L406 - Could you comment on the representativeness of these measurements for modeling efforts? I.e., would you expect these [INP] to change for different seasons based on Rosinski's work? What size bins would you expect to reach cloud level and therefore what [INP]?

Technical comments:

L2 – "are referred to as ice nucleating particles (INP)." Should be: "are referred to as ice nucleating particles (INPs)."

L3 – "mid- and high-latitude oceans" – I think there is general consensus that bubble bursting at the ocean surface (regardless of latitude) is a source of aerosol

L11 – may be helpful to add the latitude here since your reference "similar latitudes"

L18 – "Biological particles were likely found to be very important" should be "Biological particles were found to be likely important"

L19 – "A variety of bacteria and fungi were identified." – identified as what?

L20 – "Although the majority are of terrestrial origin, some of them are clearly oceanic." – majority of what? What is "them"?

L95 – fix lat/lon format

Fig 1 – would be beneficial to add a scale bar to this photos

L138 – is stage one 0.18 micron or 10 micron?

L276 – Are these error bars a standard deviation? Please define in figure caption

Table 3 – does this source correspond to all of the Genus listed?

References

Bigg, E. K. (1973). Ice nucleus concentrations in remote areas. Journal of the Atmospheric Sciences, 30, 1153–1157. https://doi.org/10.1175/1520-0469(1973)030<1153:INCIRA>2.0.CO;2

---

## Author Comment (AC1) · 13 Mar 2019

**Responses to Reviewer's Comments**

We would like to thank the Reviewers for their helpful comments and suggestions, which led to an improved manuscript. Specific answers and revisions to the text and figures related to each reviewer's comments are given below in red bold text.

**Reviewer #1**

**General comments**

This manuscript presents data on physical, chemical and biological aerosol parameters observed during a campaign lasting two weeks at a coastal site on the Yucatan peninsula. Technically, the measurements were well done and there are little similar data from the same region. In general, the manuscript is clearly written. The data obtained with the various instruments employed are weakly related, in part because of a mismatch in sampling duration and timing between INP and other parameters (chemical and biological). The sampling duration was per sample 6 hours for INP (2-3 samples a day, morning and afternoon), 48 hours for chemical components 5 min for bacteria and fungi (4 samples a day, in the morning). Further, no link can be made between the cultured microorganisms identified (Tab. 2) and ice nucleation activity, because the cultured organisms were not tested for ice nucleation activity. Much of the relations discussed between INP and other parameters are speculation supported through reference to other literature, without the current study adding substantial new evidence in support of it. Because of that, I would like to encourage the authors to put more effort into relating the different parameters in a way that each parameter can tell us more than its individual story. To start, you could try to combine data from the optical particle counter with size resolved INP concentrations to tell for different size classes the fraction of particles that are ice nucleation active (e.g. what was the ratio INP/aerosol particle in the different MOUDI size classes? How did the ratio change during the passage of the cold front?).

A: We acknowledge the reviewer's careful review and concrete suggestions that have led to an improved manuscript. In particular, the revised version now includes correlations between the chemical composition and the [INP]. Additionally, the [INP] was correlated with the particle number concentration from the LasAir to obtain more robust conclusions (Figure 7 in the revised version).

**The influence of cold fronts resulted in a higher fraction of particles acting as INP, especially for particles between 1.0 $\mu$ m and 5.0 $\mu$ m in size.**

**Specific comments**

Page 5, line 136-138: Measurements of INP concentrations with the cold cell need to be described in more detail. At least their principle should be clear to reader without having to look up the paper by Mason et al. (2015a).

A: This section has now been expanded to provide additional information of the experimental setup. The following text was added:

Lines 147-154: "The cold cell-microscope system used here is the same one used in previous studies (Mason et al., 2015a,b, 2016; DeMott et al., 2016; Si et al., 2018). The following steps encompass the analysis: i) The samples collected on glass cover slips were placed in the cold cell at room temperature, ii) The cold cell was isolated and kept at 0 °C, while humid air (RH>100 %) was injected into the cell to induce liquid droplets' formation by water vapor condensation; iii) Dry air (N2) was then injected into the cold cell to prevent the newly formed droplets from touching. This is a key step to minimize the probability of liquid droplets freezing by contact; and iv) Once droplets' sizes and thermodynamic conditions were stable, the cold cell was closed."

Lines 159-174: "The temperature at which each droplet froze was determined by analyzing the video from the CCD camera (XC-ST50, Sony) connected to the microscope and the data reported by the resistance temperature detector (RTD) located at the center of the cold cell with a ±0.2 °C uncertainty (Mason et al., 2015b). Homogeneous freezing experiments were performed on laboratory blanks exposed during the preparation of the MOUDI, while heterogeneous freezing experiments were run on ambient particles deposited on the glass cover slips (Figure S1). The [INP] was calculated using the following expression:

$$[INPs(T)] = -ln\left(\frac{N_u(T)}{N_o}\right) \cdot \left(\frac{A_{deposit}}{A_{DFT}V}\right) \cdot N_o \cdot f_{ne} \cdot f_{nu,0.25-0.10\ mm} \cdot f_{nu,1\ mm},\tag{1}$$

where  $N_u(T)$  is the number of unfrozen droplets at temperature *T*,  $N_o$  the total number of droplets,  $A_{deposit}$  the total area of the aerosol deposit on the hydrophobic glass cover slip,  $A_{DFT}$  the area of the hydrophobic glass cover slip analyzed in the DFT experiments, *V* the total volume of air sampled,  $f_{ne}$  a correction factor to account for uncertainty associated with the number of nucleation events in each experiment,  $f_{nu, 0.25-0.10 \text{ mm}}$  and  $f_{nu,1 \text{ mm}}$  a non-uniformity factor which corrects for aerosol deposit inhomogeneity at a scale of 0.25 - 0.10 mm, and 1 mm, respectively (Mason et al., 2015a). We refer the readers to Mason et al. (2015a) and Mason et al. (2015b) for more details of the MOUDI-DFT operational principle."

Fig. 3 duplicates the time series measured by the CPC, which is already shown in Fig. 2. Combine Fig. 2 and Fig. 3, and make the time series of wind speed and direction (now panel B in Fig. 3) the top panel of the combined Figure because wind is the factor driving the aerosol concentrations, so logically this factor should come first.

**A: Thank you for the suggestion. Figures 2 and 3 have now been combined.**

Figure 4: I would like to see the data of the present study as points, not just as a shaded area, where I can not see by how many points a particular part of the area defined. In particular, I am curious to see how many points support the very high [INP] at temperatures above -10 °C. I suggest to revise the Figure in a way that the data taken from Kanji et al (2017) are shown as shaded areas only (no points) and the data of the present study are superimposed on this background as points (perhaps use different symbols for data obtained during the passage of cold fronts).

A: We agree with the reviewer's suggestion and the figure was completely modified as shown below. This new figure shows the average [INP] as a function of MOUDI stage and also the combined concentration (as grey triangles). The large amount of data from Kanji et al. (2017)

were removed in the revised figure and the revised figure only includes the coastal/oceanic data from DeMott et al. (2016), Welti et al. (2018) and Irish et al. (2019) for comparison.

Figure R1. Summary of the INP concentrations as a function of temperature and particle size (solid symbols). Total [INP] are represented by the grey triangles, whereas the brown asterisks, light blue dotted lines, and purple stars are literature data from DeMott et al. (2016), Welti et al. (2018), and Irish et al. (2019), respectively. The upper and lower detection limits of the MOUDI-DFT are 30 L-1 and 0.01 L-1, respectively.

The individual INP scans that corresponded to the shaded area in the old Figure 4 is now presented in the supplementary material (Fig S6). In this improved presentation of the data (Figure 4 and Figure S6) it is clear that particles with sizes between 1.0  $\mu$ m and 1.8  $\mu$ m are the most active for temperature above -15°C.

Page 8, first line (and page 11, line 344): The aerosol number concentrations are reported as mean plus-minus one standard deviation, assuming a normal distribution of values. Although this is common practice, it is not correct because aerosol number concentrations have a log-normal distribution. I strongly encourage the authors to apply the less common, but correct metrics (median and multiplicative standard deviation) as explained in Limpert et al. (2001; BioScience, 51, 341-352, freely available at: https://stat.ethz.ch/~stahel/lognormal/bioscience.pdf). Why perpetuate a common mistake?

A: We thank the reviewer for this suggestion. The values were adjusted taking into account the log-normal distribution. Lines 262-264: "with geometric mean concentrations (assuming log-normal distributions, Limpert et al. (2001)) for the entire sampling period of 758.51 x/ 1.76 cm-3 and 1.00 x/ 1.37 cm-3, respectively."

The Conclusion section is mostly a summary of the Results and Discussion section. It should go further than that.

A: This section has now been modified, given the new analysis performed, and also includes better explanations/conclusions for our results.

**Technical corrections**

Title: Perhaps replace "importance" with "contribution"

**A: The title was changed to "Ice Nucleating Particles in a Coastal Tropical Site"**

Page 1, last line: The statement "Biological particles were likely found to be very important. . ." does not make sense to me. Do you mean "Biological particles could potentially be very important."

**A: This sentence was modified.**

Page 2, line 37: "Given the potential INP role of a variety of aerosol particles . . ." do you mean "Given the potential role of a variety of aerosol particles as INPs. . ."

Page 3, line 88: "presents", not "present"

**A: Fixed.**

Page 3, last line: Why "importance" and not simply "potential relevance"? The word "importance" is a premature value judgement here, at the end of the introduction section.

**A: "Importance" was replaced by "potential relevance".**

Page 5, line 128: I would turn the order of the cut-sizes the other way round, so that it follows the logic of the instrument (i.e. 10 um, 5.6 um, ...0.18 um).

**A: As suggested by the reviewer, the order of the cut-sizes was changed.**

Page 6, line162: delete "with"; line 165: replace "0" (zero) in "Na2CO3" with "O" (capital "o").

**A: "with" was deleted and Na2CO3 was fixed.**

Page 8, line 260: change to "At temperatures. . ." (plural).

**A: Fixed.**

Page 9, line 273: "Saclay", not "Saclary".

**A: Fixed.**

Page 10, line 311: change "elements/cations/ions" to "elements and ions are sodium and chlorine, respectively chloride"; line 322: why "elements/cations/ions" and not just "elements and ions", cations are ions.

**A: Agreed and modified in the revised manuscript.**

Page 11, line 353: "temporal mismatch of the data", not "uncertainty in the analysis".

**A: This paragraph was modified and the sentence was deleted in the revised version.**

**Reviewer #3**

**General Comments:**

This article discusses INP measurements carried out at Sisal on the coast of the Gulf of Mexico and attempts to identify the sources of INP. They have identified the biological particles from the tropical ocean as the source of INP at measurements site when the wind direction is from the GoM. Considering the importance of the INPs and lack of measurements available over the in tropical latitudes, this article makes a compelling case for the publication. The article is well written although interpretation of measurement can be better. Agreeing with the comment by an anonymous referee, who has commented on the article exhaustively, the authors need to explain the method of INP measurement in brief since that is the main thrust of the article. The lack of correlation between size bin and INP at different temperatures in Fig. 6 warrants detail analysis and in-depth discussion. I would encourage the author to normalize the INP concentration with the total number of particles in respective size bins. Similarly, the chemistry can be analyzed for bins to segregate the contribution of particles of different chemical composition.

A: The correlation between the [INP] and particle size at -20 °C and -25 °C in this study is similar to the results found by Mason et al. (2015b) in the west coast of Canada. However, our results differ at -15 °C and -30 °C and it can partly be attributed to differences in airmass history. This is now acknowledged in the text as follows:

Lines 361-369: "The discrepancies between the present results and those from Mason et al. (2015b) at -15°C and -30°C could be explained by differences in airmass history. Although both studies were conducted at coastal locations, the back-trajectories from the present study indicate that during ``normal'' days (i.e., 70% of the time) the sampled air masses had a significant continental contribution (Fig. S2). In contrast, air masses were mostly maritime in the Mason et al. (2015b) study. Also, it is important to note that although the cold air crossed the GoM before reaching Sisal, both the US Central Plains and the GoM were likely sources for the aerosol particles present in those cold air masses (Figs. S2B-C and S5)."

The experimental setup to determine the INP concentrations was extended (see above answer to reviewer #1). Additionally, the fraction of aerosol particles acting as INPs as a function of size is now shown in Figure 7.

When correlating the chemical composition and [INP] we found correlation between the average composition per sample with average [INP] per samples as shown in Tables 2 and S1. However, when the correlation is calculated as a function of particle size (per stages), the highest r2 obtained was just 0.22. It is important to note that these correlations could be misleading given that the XRF reports the mass of the elements (ng m-3) while the MOUDI-DFT reports the number of INPs (# L-1). It is well known that submicron particles are in high concentrations in comparison to supermicron particles; however, their mass is usually very low, in some cases at or below the detection limit of the analytical technique.

**Technical comments**

Line 10: should it be tropical instead of topical(?)

**A: Fixed.**

Line 35: It is encouraged that the author cites the original literature along with the newer cross-references. e.g. Kanji et al. (2017) cite other older references.

A: The text was modified to include references to the original literature as follows. Lines 34-38: "Murray et al. (2012) and Ladino et al. (2013) have suggested that contact freezing and immersion freezing are the most efficient mechanisms leading to ice nucleation in clouds; however, the atmospheric relevance of contact freezing is still under debate (Hobbs and Atkinson, 1976; Ansmann et al., 2005; Cui et al., 2006; Phillips et al., 2007; Seifert et al., 2011; Kanji et al., 2017)."

Line 41: Using the word "most important" may not be a good idea.

A: As suggested by the reviewer, "the most important" was replaced by "...has been recognized as a very important INP...".

Line 110: It would be nice if "wet" is quantified in terms of relative humidity if the measurement is available.

A: The original manuscript (line 133) mentioned that the ambient RH during the field campaign was typically above 67%. Nevertheless, the RH was added to this sentence for clarity and it now reads. Line 114: "and only wet aerosol particles were sampled (mean RH=69 %)".

Line 115: Figure 2 shows three types of time series for 3 different cut off of the particles sizes. Since the instruments used employ different principles of measurements, it would be appropriate to explain the instrument principle in brief.

A: The following text was added. Lines 122-127. "In the CPC, the size of the aerosol particles is increased in a heated saturator/cooled condenser system prior to their detection. The particles grown are directed towards a laser beam and the dispersed light is collected by a photodetector that convert it to particle concentration. Similar to the CPC, aerosol particles in the LasAir are counted by passing them through a laser beam (without any prior treatment). Based on the pulses (or voltage) and their amplitude the dispersed light by the particles is then converted to particle concentration and size."

Line 235: It should be Fig. 3 instead of Figure 3.

A: Fixed.

**Reviewer #4**

This manuscript presents the properties of ambient particles and INPs in the tropical latitudes using several instruments and sampling techniques, in order to provide a comprehensive analysis of the chemistry and biological properties of the INPs. Characterizing INP concentration around the world and understanding the contribution of bioaerosols are important, and therefore I find this study a valuable contribution, especially since there is little data from tropical latitudes.

The introduction part is good and the manuscript is well written. However, at some parts of the manuscript I felt the need for more details, as in the ice nucleation part (see below). Also, I do not think that the title reflects the findings well. I think it confuses the reader to think there is some new evidence on biological INPs, however, it was not proved that the bioaerosols are responsible for the ice nucleation activity that was detected, and I personally still not convinced. Thus, I would recommend changing the title to a more suitable one, or provide the proof for the importance of the biological particles to the INP concentrations.

**A: The requested details were added to the introduction and the title was modified to better reflect the results and conclusions. The revised title of the study is now: "Ice Nucleating Particles in a Coastal Tropical Site"**

I think that the parts related to the freezing experiments and results, should be revised and clarified. There is not enough technical details about the method, such as the instrumentation or the temperature uncertainty, and what are the limitation of the analysis that was done.

**A: More information regarding the ice nucleation experimental setup and uncertainties was added to the revised manuscript:**

Lines 147-154: "The cold cell-microscope system used here is the same one used in previous studies (Mason et al., 2015a,b, 2016; DeMott et al., 2016; Si et al., 2018). The following steps encompass the analysis: i) The samples collected on glass cover slips were placed in the cold cell at room temperature, ii) The cold cell was isolated and kept at 0 °C, while humid air (RH>100 %) was injected into the cell to induce liquid droplets' formation by water vapor condensation; iii) Dry air (N2) was then injected into the cold cell to prevent the newly formed droplets from touching. This is a key step to minimize the probability of liquid droplets freezing by contact; and iv) Once droplets' sizes and thermodynamic conditions were stable, the cold cell was closed."

Lines 159-174: "The temperature at which each droplet froze was determined by analyzing the video from the CCD camera (XC-ST50, Sony) connected to the microscope and the data reported by the resistance temperature detector (RTD) located at the center of the cold cell with a ±0.2 °C uncertainty (Mason et al., 2015b). Homogeneous freezing experiments were performed on laboratory blanks exposed during the preparation of the MOUDI, while heterogeneous freezing experiments were run on ambient particles deposited on the glass cover slips (Figure S1). The [INP] was calculated using the following expression:

$$[INPs(T)] = -ln\left(\frac{N_u(T)}{N_o}\right) \cdot \left(\frac{A_{deposit}}{A_{DFT}V}\right) \cdot N_o \cdot f_{ne} \cdot f_{nu,0.25-0.10\ mm} \cdot f_{nu,1\ mm},\tag{1}$$

where  $N_u(T)$  is the number of unfrozen droplets at temperature *T*,  $N_o$  the total number of droplets,  $A_{deposit}$  the total area of the aerosol deposit on the hydrophobic glass cover slip,  $A_{DFT}$  the area of the hydrophobic glass cover slip analyzed in the DFT experiments, *V* the total volume of air sampled,  $f_{ne}$  a correction factor to account for uncertainty associated with the number of nucleation events in each experiment,  $f_{nu, 0.25-0.10 \text{ mm}}$  and  $f_{nu,1 \text{ mm}}$  a non-uniformity factor which corrects for aerosol deposit inhomogeneity at a scale of 0.25 - 0.10 mm, and 1 mm, respectively (Mason et al., 2015a). We refer the readers to Mason et al. (2015a) and Mason et al. (2015b) for more details of the MOUDI-DFT operational principle."

Also - there are no error bars (or any error analysis) throughout the manuscript, which is very surprising, and must be done before resubmission.

A: The original manuscript included uncertainty bars in Figures 5 and 8; uncertainty bars have now been added to Figure 4, Figure 6, Figure 7, and Figure S4.

Specific comments:

Line 10: I guess should be "tropical" and not topical

**A: Fixed.**

Line 13: for -> from

**A: Fixed.**

Line 29: is -> was

**A: Fixed.**

Line 40: I suggest to detail how the concentration affect on water drop formation

A: Is the reviewer referring to the following statement: "Mineral dust has been recognized as the most important INP on a global scale due to their good ice nucleating abilities and their elevated concentrations in the troposphere"? If yes, we are referring to the INPs abilities of mineral dust and not to their CCN abilities. If there are more mineral dust particles, the likelihood that some of them can act as INPs is higher.

Line 88: present -> presents

**A: Fixed.**

Line 110: What is wet?

A: The original manuscript (line 133) mentioned that the ambient RH during the field campaign was typically above 67%. Nevertheless, the RH was added to this sentence for clarity and it now reads: Line 114: "and only wet aerosol particles were sampled (mean RH=69 %)".

Line 235: if possible, please explain here how was the cold front determined, and send to the supplementary figures.

A: Figure 2A (Fig 3B in the original manuscript) shows the time series of the wind speed during the field campaign. Cold fronts are associated with significant changes in wind speed, wind

direction, and temperature. We use here changes in the wind speed data, which were more pronounced than in direction. Additionally, surface maps from NOAA were used to confirm the passage of the cold fronts as shown in Figure S3.

Line 250: Please explain why do you believe that there is no significant difference and that during front did not changed much. To me it seems that their slope is different and that the concentrations are also differ.

A: We agree with the reviewer. Both the slopes and concentrations are different, especially for submicron particles. The text was modified as follows. Lines 286-289: "As for the total aerosol concentration (Figure 2), the number size distributions of the aerosol particles larger than 300 nm were also impacted by the cold fronts. For example, the concentration of particles smaller than 5.0  $\mu$ m was lower during the passage of the cold fronts (Figure S4)."

Line 265: I suggest rephrasing this sentence; especially the word important, it does not fit here.

A: The text was changed as follows. Lines 316-317: "for typical atmospheric concentrations of mineral dust, ice nucleation at these temperatures seems to be of secondary importance".

Line 332: I suggest removing or rephrasing this paragraph; it is a speculation that is not accurate.

A: The sentence was modified to address the new results shown in Table 2.

Line 353: Can you detail what is the "uncertainties of analysis"

A: This was a grammatical mistake. Our intention was to acknowledge that a direct correlation between the MOUDI and biosampler data was not completely fair given that the different sampling time introduces uncertainty. The text was revised as follows. Lines 424-425: "This poor correlation can be in part due to the different sampling time of the MOUDI and the biosamplers".

Line 451: fix reference of Chen et al. 2011.

A: We are sorry, but we could not find any mistake in this reference. Can the reviewer please let us known what is the problem here?

Chen, S.-C., Tsai, C.-J., Chen, H.-D., Huang, C.-Y., and Roam, G.-D.: The influence of relative humidity on nanoparticle concentration and particle mass distribution measurements by the MOUDI, Aerosol Sci. Technol., 45, 596–603, doi:10.1080/02786826.2010.551557, 2011.

Fig.2 - I suggest to write "entire" instead of "whole"

A: Changed.

Fig 2. Is seems to me that there is daily cycle of the aerosols? For example seen clearly in days 27-28.1.17.

A: We agree with the reviewer. Thanks for pointing this out. This is acknowledged in the revised manuscript. Lines 264-265: "From the CPC data shown in Figure 2A, there seems to be a daily cycle with most of the highest concentration taking place between 7 h and 12 h (local time)"

Figure S1. I would consider replacing the use of brackets in this caption. It was not clear to me at first what I see there.

A: The left), middle), and right) labels were replaced by A), B), and C) for clarity.

Figure S3. Why there are no error bars? That way it will be easier to understand if the two distribution are actually similar.

A: Uncertainty bars were added to three of the seven size distributions only for readability. When all seven size distribution have uncertainty bars, the figure is more difficult to understand.

**Reviewer #5**

"The Importance of Biological Particles to the Ice Nucleating Particle Concentrations in a Coastal tropical Site" by Ladino et al. describes efforts to characterize the INP population and biological particles at a tropical site. These data are valuable to the community due to a lack of data in such environments. My major comment is that more information on the INP measurement detection limits, blanks, and uncertainties is needed because the paper heavily relies on these data.

**A: More information related to the experimental setup and its uncertainties was added to revised manuscript:**

Lines 147-154: "The cold cell-microscope system used here is the same one used in previous studies (Mason et al., 2015a,b, 2016; DeMott et al., 2016; Si et al., 2018). The following steps encompass the analysis: i) The samples collected on glass cover slips were placed in the cold cell at room temperature, ii) The cold cell was isolated and kept at 0 °C, while humid air (RH>100 %) was injected into the cell to induce liquid droplets' formation by water vapor condensation; iii) Dry air (N2) was then injected into the cold cell to prevent the newly formed droplets from touching. This is a key step to minimize the probability of liquid droplets freezing by contact; and iv) Once droplets' sizes and thermodynamic conditions were stable, the cold cell was closed."

Lines 159-174: "The temperature at which each droplet froze was determined by analyzing the video from the CCD camera (XC-ST50, Sony) connected to the microscope and the data reported by the resistance temperature detector (RTD) located at the center of the cold cell with a ±0.2 °C uncertainty (Mason et al., 2015b). Homogeneous freezing experiments were performed on laboratory blanks exposed during the preparation of the MOUDI, while heterogeneous freezing experiments were run on ambient particles deposited on the glass cover slips (Figure S1). The [INP] was calculated using the following expression:

$$[INPs(T)] = -ln\left(\frac{N_u(T)}{N_o}\right) \cdot \left(\frac{A_{deposit}}{A_{DFT}V}\right) \cdot N_o \cdot f_{ne} \cdot f_{nu,0.25-0.10\ mm} \cdot f_{nu,1\ mm},\tag{1}$$

where  $N_u(T)$  is the number of unfrozen droplets at temperature *T*,  $N_o$  the total number of droplets,  $A_{deposit}$  the total area of the aerosol deposit on the hydrophobic glass cover slip,  $A_{DFT}$  the area of the hydrophobic glass cover slip analyzed in the DFT experiments, *V* the total volume of air sampled,  $f_{ne}$  a correction factor to account for uncertainty associated with the number of nucleation events in each experiment,  $f_{nu, 0.25-0.10 \text{ mm}}$  and  $f_{nu,1 \text{ mm}}$  a non-uniformity factor which corrects for aerosol deposit inhomogeneity at a scale of 0.25 - 0.10 mm, and 1 mm, respectively (Mason et al., 2015a). We refer the readers to Mason et al. (2015a) and Mason et al. (2015b) for more details of the MOUDI-DFT operational principle."

I have several other concerns in the interpretation of results, described below. In general, the authors motivate the study by describing the need to characterize marine INP sources in the tropics ("Very few studies to sample INPs have been carried out in tropical latitudes, and there is a need to evaluate their availability to understand the potential role that marine aerosol may play in the hydrological cycle of tropical regions"), but I am not convinced the method deployed can measure [INP] for a remote region and I think the concentrations reported could not possibly be explained by marine aerosol.

A: As shown in Mason et al. (2016), this method has been successfully used in different remote regions, such as Alert (Canada), Labrador Sea, Ucluelet (Canada). Additionally, it is important to mention that this method has been correlated and validated with other ice nucleation instrumentation such as the Colorado State University-Continuous Flow Diffusion Chamber (CSU-CFDC), obtaining comparable results (DeMott et al. 2016). There are more than eight papers already published using the same technique used in this study. Additionally, the [INP] reported in this study (Figure 5) is on the same order (albeit slightly higher), than those reported in other coastal/marine locations using the same technique (Mason et al. 2015a).

Finally, as illustrated in Figure 8 the chemical analysis performed by two different analytical techniques (i.e., HPLC and XRF) shows the presence of Na and Cl in significant concentrations, confirming that the air masses sampled were clearly maritime.

Furthermore, additional analysis carried out on samples collected at the same site in July (not relevant for the manuscript under review, and not yet published), is presented here to address the reviewer's concern about the maritime origin. The Table below shows particle chemical composition sampled in Sisal under the influence of Saharan dust (July 2018), with a dominant signature in Si, Fe, and Al, instead of the Na and Cl dominant in the wintertime samples discussed in the present study.

Table R1. Comparison of the aerosol chemical composition in Sisal during winter 2017 and summer 2018.

| Element | XRF concentration
in Winter 2017 (ng
m -3 ) | XRF concentration
in Summer 2018
(ng m -3 ) |
|---------|--------------------------------------------------------------|--------------------------------------------------------------|
| Na      | 1.83                                                         | 0.38                                                         |
| CI      | 5.49                                                         | 0.23                                                         |
| AI      | 0.11                                                         | 1.00                                                         |
| Si      | 0.22                                                         | 1.72                                                         |
| Fe      | 0.21                                                         | 0.96                                                         |

The new analysis performed (Table 2 in the revised manuscript) suggests that mineral dust particles could be responsible for INPs between -20 °C and -30 °C. However, this is not the case for INPs measured at -15 °C. This is now discussed in the revised manuscript.

**Bibliography cited in this reply:**

- DeMott, P. J., Hill, T. C., McCluskey, C. S., Prather, K. A., Collins, D. B., Sullivan, R. C., Ruppel, M. J., Mason, R. H., Irish, V. E., Lee, T., et al.: Sea spray aerosol as a unique source of ice nucleating particles, P. Natl. Acad. Sci., 113, 5797–5803, doi:10.1073/pnas.1514034112, 2016.
- Mason, R., Chou, C., McCluskey, C., Levin, E., Schiller, C., Hill, T., Huffman, J., DeMott, P., and Bertram, A.: The micro-orifice uniform deposit impactor-droplet freezing technique (MOUDI-DFT) for measuring concentrations of ice nucleating particles as a function of size:

improvements and initial validation, Atmos. Meas. Tech., 8, 2449–2462, doi:10.5194/amt-8-2449-2015, 2015a.

Mason, R., Si, M., Chou, C., Irish, V., Dickie, R., Elizondo, P., Wong, R., Brintnell, M., Elsasser, M., Lassar, W., et al.: Size-resolved measurements of ice-nucleating particles at six locations in North America and one in Europe, Atmos. Chem. Phys., 16, 1637–1651, doi:10.5194/acp-16-1637-2016, 2016.

Overall, I think there are a few things that need to be clearly stated and supported consistently throughout: 1) What were the [INP] and their variability (and their detection limits)?

A: This information is now provided in the revised Figure 4 and Figure S5. The following text was added to Figure captions: "The upper and lower detection limits of the MOUDI-DFT are 30 L-1 and 0.01 L-1, respectively."

2) What size range corresponded to the highest [INP]?

A: This information is now shown in new Figure 4. The following text was added. Lines 311-313: "The "high" [INP] found at -15 °C can be explained in part by the very efficient INPs shown in Figure S6, with sizes ranging from 1.0  $\mu$ m to 1.8  $\mu$ m. However, it is important to note that particles with diameters between 1.8  $\mu$ m and 10  $\mu$ m also contribute to the total [INP] at warm temperatures."

3) what meteorological conditions, air mass histories (back trajectory and PCR results) corresponded to the highest [INP]?

A: Figures 5 and 7 indicate that the cold air masses behind the cold fronts show higher [INP].

4) What is the hypothesized origin of these very high [INP] and biological particles? If the authors can build up the results discussions around some clear points, I think it will be easier to follow along.

A: Figure 4, 6, and 7 suggests that the more efficient INPs are those supermicron in size. Additionally, based on the results summarized in Tables 2 and 3, and given that at -15 °C mineral dust is not the likely source of the measured INPs, we hypothesize that they are biological (both continental and marine). However, given that the [INP] under the influence of cold fronts is higher, it may be more likely that the efficient biological INPs are of marine origin.

The following text was added. Lines 482-487: "Based on the large [INPs] above -15 °C, the supermicron size of 90 % of the INPs, the presence of marine biological particles in the cold air masses those of which showed the highest [INP], in addition to the poor correlation shown by the mineral dust tracers with the [INP], we hypothesize that the likely source of the INPs measured at high temperatures in Sisal are biological particles. Therefore, our results suggest that continental and maritime biological particles could play an important role in ice cloud formation and precipitation development in the Yucatan peninsula."

**Comments:**

Abstract: Should mention the freezing mode and INP temperature range measured during this study in abstract.

A: This is now stated in the revised version. Lines 12-13: "Aerosol particles sampled in Sisal are shown to be very efficient INPs in the immersion freezing mode, with onset freezing temperatures..."

L18 – I think a better way to say this is similar to how it was stated in the results section, something like "The high concentrations of INPs at warmer ice nucleation temperatures (T >-15C) and the supermicron size of the INPs suggest that biological particles may have been a significant contributor to the INP population in Sisal during this study".

A: We thank the reviewer for the suggestion and the text was revised accordingly.

L55: The modeling studies listed determined specific regions where oceanic sources dominated the INP population due to an absence of other types, like mineral dust.

A: The sentence was modified as follows for clarity. Lines 56-60: "Marine organic matter, likely of biological origin, has been suggested to be an important oceanic source of INPs in the southern oceans, north Atlantic, and north Pacific (Burrows et al., 2013; Yun and Penner, 2013; Wilson et al., 2015; Vergara-Temprado et al., 2017); however, the maritime source suggestion was made with little or no data from tropical latitudes".

L58: Bigg (1973) was the first to report such a study (his data are used in Schnell and Vali, 1975)."

A: Thank you for pointing this out. The citation to the Bigg (1973) study was added to the revised manuscript.

L67 – the range may also be from different species, right?

A: You are correct. However, this is a very general statement that aims to introduce the large [INP] range in marine environments. We do not see the need to modify the statement.

L142 – please provide information on the measurement detection limit and how measurement uncertainty was determined. Where there blanks collected and how were these accounted for?

A: See the first answer to reviewer #5 above. Homogeneous freezing experiments were performed on blanks. The blanks refer to clean glass covered slips exposed while mounting and preparing the MOUDI in the lab during the field campaign. The figure R2 included below was added to the supplementary material and it shows examples of activation scans for blanks and ambient particles.

---

## Author Response (AR2)

**Responses to Reviewer's and Editor Comments**

**We would like to thank the Reviewers and the Editor for their valuable additional comments and suggestions. Specific answers and revisions to the text and figures related to each comment are given below in red bold text.**

**Editor**

• Line 3, page 1: Why only "mid- and high-latitude" oceans? There is no reason why other oceans shouldn't release material that can act as INP. Suggest to delete "mid- and high-latitude"

**A: Deleted.**

• Line 5, page 1: Delete "the" before oceans.

**A: Deleted.**

• Line 37, page 2: Can a reference from 1976 still be part of a current debate? Suggest to reformulate this sentence.

**A: The sentence was modified as follows (Lines 36-37): "however, the atmospheric relevance of contact freezing is still unclear given the contradictory results".**

• Line 99, page 4 and throughout the manuscript: The geographic abbreviations (N, W) should not be in italics (remove the Latex mathmode).

**A: Fixed.**

• Line 109, page 4: For completeness, add "(arithmetic mean ±standard deviation)"

**A: Added.**

• Line 129, page 5: Please mention that these are "optical diameters". Is the cut-size the upper, lower or mid-bin size? I guess the last bin counts all particles larger than 25 micron?

**A: "Optical diameter" was added.**

• Line 137, page 5: Please precise the size ranges of the different stages. I guess the first one is ">10 micrometer", the second one "10-5.6 micrometer", etc.? See also line 156.

**A: The following text was added to the revised manuscript (Lines 139-140): "The particle size range for each MOUDI stage are given in Table S1". Table S1 was added to the Supplementary Material and the size range was added to Line 156.**

**Table S1.** Particle size range for each MOUDI stage.

| MOUDI stage | Size range ($\mu$m) |
|---|---|
| 1 | >10.0 |
| 2 | 5.6 – 10.0 |
| 3 | 3.2 – 5.6 |
| 4 | 1.8 – 3.2 |
| 5 | 1.0 – 1.8 |
| 6 | 0.56 – 1.0 |
| 7 | 0.32 – 0.56 |
| 8 | 0.18 – 0.32 |

• Equation 1, line 166 and later in the manuscript: The subscripts in the equation or of variables in generals should not be in italics. Add "\rm" in the parenthesis (e.g. A_{\rm deposit}). Instead of "ln" use "\ln" since this is a function.

**A: Fixed.**

• Line 254, 255 and also later in the manuscript: The capitalization of the section headings suddenly changes compared to previous headings … Please harmonize.

**A: Fixed.**

• Line 258, page 8 and throughout the manuscript: Within the sentence the word figure should be "Fig." while "Figure" when it is at the beginning of the sentence. Please have a look at https://www.atmospheric-chemistry-and-physics.net/for_authors/manuscript_preparation.html and carefully fulfil the manuscript preparation requirements.

**A: Fixed here and along the manuscript.**

• Line 273, page 9: "than" -> "that", add "the" before "cold"

**A: Added.**

• Line 276, page 9: "Hysplit" -> "HYSPLIT", move "from the measurement site" to the front "Back-trajectories from the measurement site …"

**A: Fixed.**

• Line 296, page 9: I suggest to remove the parenthesis "([INP])" from the heading title, it should be defined within the text (as has been done before).

**A: Deleted.**

• Line 400, page 12: O'Dowd

**A: Fixed.**

• Line 415, page 13: Good correlation with what exactly? INP concentration?

**A: The sentence was modified as follows (Lines 434-435): "The bacteria and fungi concentrations showed a relatively good correlation between each other."**

• Line 460, page 14: Add "collected" or "sampled" before "around Sisal".

**A: "collected" was added.**

• Line 466, page 14: Maybe better: "… using the same type of INP counter." I would also start the issue of the influence of the cold fronts in a new sentence.

**A:  The sentence text was modified as follows (Lines 485-487): "at other locations studied using the same type of INP counter. The higher INP concentrations were observed especially under the influence of cold fronts."**

• Line 468, "than" -> "that"

**A: Fixed.**

• Line 473: Better: "However, the concentrations of Al, Si, …"

**A: Changed.**

• Line 476: "magnesium" -> "Mg" (to be consistent)

**A: Fixed.**

• Page 14 & 15: Large parts of the conclusions, especially the last three paragraphs, are more a discussion than a precisely formulated conclusion and should be moved to Sect. 3 (Results and Discussion).

**A: The following text was moved from the "Conclusions" to the "Results and discussion section" to improve the readability of the manuscript:**

**Lines 413-417: "From the correlation of the [INP] and the aerosol chemical composition at -15 °C, Mg was the only element showing a correlation that is statistically significant at the 95 % confidence interval (p<0.05). Although Mg can be found in mineral dust particles in low percentages, it can also be found in marine environments linked to sea spray aerosol (e.g., Savoie and Prospero (1980); Andreae (1982); Casillas-Ituarte et al. (2010))."**

**Lines 429-430: "As stated by Islebe et al. (2015) both bacteria and fungi need to be properly documented in the peninsula and the GoM to fully understand their regional importance."**

**Lines 423-426: "Efficient INPs such as those measured in Sisal could be very important for cloud glaciation. Additionally, they can trigger ice multiplication or secondary ice formation at such high temperatures via the Hallett-Mossop mechanism (Hallett and Mossop, 1974; Field et al., 2017) and impact precipitation formation."**

• A statement on the data availability is missing (see https://www.atmospheric-chemistry-and-physics.net/about/data_policy.html).

**A: The following text was added (Line 517): "*Data availability*. Data are available upon request to the corresponding author."**

• Table 1: Also add the acronyms for HLPC and XRF.

**A: Added.**

• Table 2: Please add a column showing how many sample points are being compared here.

**A: Given that the number of points is the same for every temperature we think that adding a column is not necessary. The number of points is now added to the Table's caption.**

**Reviewer #1:**

The authors have improved their manuscript according to recommendations, including the representation of aerosol concentrations in terms of geometric mean and multiplicative standard deviation (i.e. lines 260-264).

**A: We thank the reviewer for its positive feedback.**

For readers not familiar with the concept, it would be helpful to slightly re-write this sentence and add the term "multiplicative standard deviation", for example in this way: "There is a large diurnal variability for the aerosol particle concentration measured by the CPC (particles > 30 nm, Fig. 2B) and the LasAir (particles >300 nm, Fig. 2C). Assuming log-normal distributions, the geometric mean concentration and multiplicative standard deviation (c.f. Limpert et al., 2001) for the entire sampling period was 758.51 x/ 1.76 cm−3 and 1.00 x/ 1.37 cm−3, respectively." In addition, authors may consider to present [INP] in the Conclusion section also in terms of geometric mean and multiplicative standard deviation. As it is now, the value of the standard deviation (assuming normal distribution) is larger than the mean (at -15 °C and -20 °C), suggesting there were instances with negative number concentrations of INP, which does not make sense.

**A: The suggested text was added to the revised manuscript. Additionally, the following text was added to the conclusions (Lines 483-485): "(geometric mean and multiplicative standard deviation of 0.44 $^x$/ 1.77 L$^{-1}$, 1.73 $^x$/ 2.56 L$^{-1}$, and 6.20 $^x$/ 2.65 L$^{-1}$ at -15°C, -20°C, and -25°C, respectively)".**

line 257: change "aire" to ""air"
**A: Fixed.**

line 263: change "than" to "that"

**A: Fixed (We think the reviewer refers to line 273 instead of line 263).**

**Reviewer #4:**
I do not recommend on this study to be published in acp, mainly because I do not see new findings that add scientific progress. The conclusions are still relying on other literature, more than on findings from this study. I think the technical part of this study was done well, and that the resulting data is worth publishing, especially due to the fact that there is not much data on INPs from this region, however, from a manuscript that is submitted to acp, it is expected that the data analysis and conclusions will be more comprehensive and innovative, as well as to fulfil its scientific potential.

**A: We disagree with the reviewer recommendation, especially because we have properly addressed each of the 20 points listed in his/her original review. In the revised manuscript we added new information to support our conclusions: 1) correlation of the [INP] with the bulk aerosol chemical composition (Fig. 3 and S5), 2) correlation of the [INP] with the total aerosol concentration and particle size (Fig. 7). These are novel measurements for this tropical site and this is one of the few studies reporting the [INP] as a function of particle size covering such a large size range (0.18 to 10 μm). In summary, the present study not only reports the [INP], but it also provides information regarding the chemical, physical, and biological characteristics of the collected aerosol in addition to the meteorological variables. The reviewer evaluation is in contrast with the**

**other two reviewers who evaluated the revised manuscript from "Good" to "Excellent" in all three categories.**

**Reviewer #5:**

Ladino et al., have thoroughly satisfied my comments and their manuscript contributes a valuable analysis and dataset for the ice nucleation community. I think the paper should be accepted for publication after addressing the following mostly minor comments.

**A: We thank the reviewer for its positive feedback.**

L57: "important oceanic sources of INPs" suggest change to "important oceanic source of INPs"

**A: Changed.**

L105/figure1: could a label be added for Sisal for those not as familiar with the geography of this region?

**A: The location of the Center of Sisal was added to Fig. 1.**

L124: "… photodetector that convert it…" change to "…photodetector that converts it…"

**A: Changed.**

L170: can the authors expand on the correction factor (fne)? Is this determine each run or for each study? What range of factors are used? Is this a passion counting statistic uncertainty?

**A: From Demott et al. 2017. "$F_{ne}$ is a correction factor to account for the statistical uncertainty that results when only a limited number of nucleation events are observed. $f_{ne}$ was calculated following the approach given in Koop et al. (1997) using a 95 % confidence interval." $F_{ne}$ is based on Poisson counting statistics and it was calculated for each run.**

L174: I think the detection limits of the MOUDI-DFT need to be included in the methods description also (rather than just the Figure 4 caption). This is an important distinction from other measurements shown on Figure 4. That is, there may be [INP] present at Sisal that are below or above the instrument's detection limit, so it's not a very good direct comparison at all temperatures. E.g., DeMott et al., reported measurements less than 0.0001 L-1, but those are not measurable by the MOUDI-DFT.

**A: The detection limits of the MOUDI-DFT were added to the Methods (Lines 175-176).**

L257: typo: "different aire mass"

**A: Fixed.**

L263: I think there may be a formatting issue here for the mean number concentrations

**A: This is the multiplicative standard deviation format used by Limpert et al. (2001) when using a log-normal distribution. The text was modified following Reviewer´s #1 suggestion.**

L272: Any precipitation from these frontal passages? This may be another explanation for lower concentrations and suppressed diurnal cycle.

**A: We thank the reviewer for bringing this up. The following text was added (Lines 276-278): "During the passage of cold front A, precipitation events were not observed which was not the case for cold front B. This could partially explain the lower aerosol concentration observed during the passage of the cold front B in comparison to cold front A".**

L276: This is typically written in all caps (HYSPLIT)

**A: Fixed.**

L289: "..during the passage of the cold fronts" should be "…during the passage of Cold Front B" I think.

**A: The reviewer is right. This was fixed.**

Figure 3: Should the start time be 00:00 h local time? The xlabels suggest that the start time was midnight?

**A: The starting time for the XRF samples was always at 12:00 h (local time). In this case the Figure caption is correct. Given the mismatch between the length of the cold fronts and the XRF sampling time, the addition of the Cold front A and B labels on Figure 3 could confuse the readers. Therefore, the "Cold front A" and "Cold front B" labels were changed by "A" and "B" indicating that this samples were partially influenced by the passage of the cold fronts A and B. The following text was added to the Figure´s caption: "A and B indicate that those samples were partially influenced by the passage of the Cold front A and the Cold front B, respectively".**

L290: sentence contains grammatical error?

**A: The sentence was modified as follows (Lines 296-298): "the XRF analysis indicates that although there are small differences in the bulk chemical composition of the**

**aerosol particles, the overall composition is generally comparable in the presence or absence of cold fronts."**

Figure S4 – Are these 24 hour averages? Perhaps add this detail to the figure caption.

**A: Yes. This was added to the figure caption.**

L292: suggest change to: "Note, however, that this is not a completely fair comparison given that sampling time for the chemical analysis was 48 h, while sampling time for determining the influence of the cold front air masses on INP populations was on the order of 36 hours. Therefore, the periods denoted as cold fronts contain aerosol particles that may not technically correspond to cold front air masses." (or something like this)

**A: Thank you for the suggestions. This was added to the revised version (Lines 298-333).**

Figure 4: "Summary of INP concentrations" suggest change to "Summary of average INP concentrations"; what are the vertical lines for the grey points, 1 standard deviation? As a suggestion to highlight the number of samples and work that was included in this project, I would pull each of the values from Figure S6 into Figure 4 as lighter colors. There is a lot of data here, and it could be lost if all just placed in the supplemental.

**A: The figure caption was changed as suggested. However, we do not feel comfortable with the second suggestion (i.e., to add the data from Fig. S6 into Fig. 4). Adding the data from Fig. S6 into Fig. 4 will make Fig. 4 very messy and difficult to follow. Additionally, the data from Fig S6. will not add additional information to Fig. 4.**

Table 1 – could move this to supplemental I think.

**A: We could; however, we think that it is better to keep it in the main text as this information may be important for some readers.**

Figure 6. It's very difficult to see the uncertainty bars on Figure 6D. could change to black bars?

**A: The color was changed to black.**

L324: Maybe specify that Kolby is "central plains/agricultural"

**A: "An agricultural site" was added (Line 331).**

L350: "Additionally, the chemical composition of the aerosol particles collected by Rosinski et al. (1988) indicate that the air masses in the GoM in July-August were significantly influenced by mineral dust particles." – wouldn't presence of dust support Rosinski's [INP] to be higher than [INP] from this study? Later the authors mention that

the aerosol particles in cold front air masses are likely a mixture of particles from US Central Plains and the GoM – could this be a difference between these measurements and those by Rosinski?

**A: There could be different reasons for the higher [INP] found in the present study in comparison to Rosinski et al. (1988). The presence of mineral dust particles will likely result in a higher [INP]; however, this will also depend on the concentration of dust particles >500 nm, and probably the presence of biological particles. Actually, our recent results (not included in this manuscript) shows that the [INP] in Sisal during the "Saharan dust season" in July 2018 was higher than in winter (Jan-Feb 2017). We believe that differences in the particle size is a key factor. Looking at Figure 4, if we do not take into account the supermicron particles, the total [INP] is lower by one order of magnitude. The following text was added to the revised manuscript (Lines 354-355): "If supermicron particles are excluded, the [INP] at -15 °C from the present study is one order of magnitude lower (Fig. 4)".**

Figure 7 – suggested change: "influence of the cold fronts (CF)".

**A: Added.**

L405: what kind of organics? Secondary organic aerosol? Bioparticles are organic.

**A: Secondary organic aerosol was added (Line 418).**